# Proteomic and metabolomic profiling of extracellular vesicles produced by human gut archaea

Viktoria Weinberger[1], Barbara Darnhofer[2], Himadri B. Thapa [3], Polona Mertelj[1], Régis Stentz[4], Emily Jones[4], Gerlinde Grabmann [5], Rokhsareh Mohammadzadeh[1], Tejus Shinde[1], Christina Karner [6], Jennifer Ober[7], Rokas Juodeikis [4], Dominique Pernitsch[8], Kerstin Hingerl[8], Tamara Zurabishvili[1], Christina Kumpitsch [1], Torben Kuehnast[1], Beate Rinner[6], Heimo Strohmaier[7], Dagmar Kolb[8], Kathryn Gotts[9], Thomas Weichhart [10], Thomas Köcher[5], Harald Köfeler [2], Simon R. Carding[4,11], Stefan Schild [3,12,13] & Christine Moissl-Eichinger [1,13] ✉

Gastrointestinal bacteria interact with the host and each other through various mechanisms, including the production of extracellular vesicles (EVs). However, the composition and potential roles of EVs released by gut archaea are poorly understood. Here, we study EVs produced by four strains of human gut-derived methanogenic archaea: *Methanobrevibacter smithii* ALI, *M. smithii* GRAZ-2, *M. intestini*, and *Methanosphaera stadtmanae*. The size (~130 nm) and morphology of these EVs are comparable to those of bacterial EVs. Proteomic and metabolomic analyses reveal that the archaeal EVs are enriched in putative adhesins or adhesin-like proteins, free glutamic and aspartic acid, and choline glycerophosphate. The archaeal EVs are taken up by macrophages in vitro and elicit species-specific responses in immune and epithelial cell lines, including production of chemokines such as CXCL9, CXCL11, and CX3CL1. The EVs produced by *M. intestini* strongly induce pro-inflammatory cytokine IL-8 in epithelial cells. Future work should examine whether archaeal EVs play roles in the interactions of archaea with other gut microbes and with the host.

All organisms have evolved various signaling mechanisms to convey crucial biological information across cells, tissues, and organs[1–3]. Among these mechanisms are extracellular vesicles (EVs), which are small membrane-bound spherical particles produced and released by cells of all three domains of life[1–3].

In the gastrointestinal tract (GIT), extracellular vesicles produced by commensal bacteria (bacterial extracellular vesicles, BEVs) mediate intra- and inter-kingdom interactions, maintaining the microbiome ecosystem and promoting interactions with the host[2].

BEVs have garnered considerable attention in recent years due to their diverse roles in intercellular communication, pathogenesis, stress tolerance, immune stimulation, and host-microbe interactions[3–7]. These small, membrane-bound structures serve as vehicles for the transport of biomolecules, such as proteins, nucleic acids, metabolites, and lipids between bacterial cells, as well as between bacteria and their host environments[4,8–12]. Understanding the mechanisms underlying BEV biogenesis, cargo loading, and their impact on microbial communities and host physiology is critical in microbiology and biomedical research[4].

BEVs are divided into different categories based on either their producing bacteria (BEVs from Gram-negative and Gram-positive bacteria) or their origin and the pathway by which they are formed

(outer membrane vesicles, outer-inner membrane vesicles, explosive membrane vesicles or cytoplasmic membrane vesicles)[13,14].

Outer membrane vesicles (OMVs) are considered as the archetypal bacterial membrane vesicles. These OMVs usually arise from a protrusion of the outer membrane including the cell envelope. Thus, they usually contain surface-associated factors, outer membrane proteins, and periplasmic content. Explosive membrane vesicles on the other hand diversify BEV composition, explaining the presence of nucleic acids and cytosolic content in vesicle samples from Gram-negative bacteria[13,14]. While OMVs are formed through blebbing, explosive membrane vesicles are generated via endolysin-induced cell lysis[13,14].

In the course of the last decade, it has become evident that BEVs of GIT-colonizing bacteria have the potential to influence essential functions of the intestine and of systemic organs after their migration to the bloodstream, thereby contributing to host health[15]. For instance, BEVs contribute to host digestion by distributing hydrolase activities across the lumen, and can potentially influence the central nervous system following migration through the gut-brain axis[16]. Additionally, BEVs can act as efficient delivery vehicles of bioactive compounds, such as toxins or modulators of host cell physiology[13,14]. BEVs are recognized and efficiently internalized by various host cells resulting in intestinal barrier changes, immunomodulation and (patho-)physiological changes[13,14]. BEVs can also act on the surrounding microbiota, promoting bacterial colonization and growth as well as protecting bacteria from antibiotics and host defense peptides[11,17,18].

Triggers for vesicle formation are manifold, including factors such as media composition, growth phase, temperature, iron and oxygen availability, as well as exposure to antibiotics and stress[13,14]. As a consequence of the diverse triggers and various origins, the vesicle preparations likely reflect a mixture of different BEV types, which could explain variable BEV functions and effects[14].

Representatives of all three domains of life, eukaryotes, bacteria, and archaea, are capable of forming extracellular vesicles[19]. Reports on archaeal vesicles are rare and restricted to extremophilic archaea, namely Thermococcales and Sulfolobales. It appears that in *Sulfolobus*, for example, vesicle formation is evolutionarily related to the eukaryotic endosomal sorting complexes required for transport (ESCRT) proteins used for the building of endosomes; however, other archaea, such as *Thermococcus* form vesicles but do lack the ESCRT complex, indicating a higher variety in vesicle formation mechanisms[19]. Vesicles formed by *Thermococcus* and other Thermococcales species serve multiple functions, primarily related to sulfur detoxification and genetic material transfer[20–23].

However, archaea not only thrive in environmental ecosystems, but are also reliable and stable constituents of the human GIT microbiome. With 1.2% relative abundance on average, *Methanobrevibacter* and *Methanosphaera* species are highly prevalent across individuals (>90%)[24,25]. Through maintaining numerous syntrophic relationships with intestinal bacteria, these archaea have the capacity to orchestrate the entire microbiome, leading to an optimized fibre degradation[26]. They also influence the host with respect to the provision of short chain fatty acids or mediate the reduction of gut motility, leading to constipation[24]. However, the mechanisms by which they interact with other microorganisms and their mode of signaling have remained unknown.

The human archaeome stays an underexplored component of the microbiome, and this knowledge gap substantially limits our understanding of how archaea contribute to human health and disease[27,28]. Unlike the extensive research on bacterial EVs, archaeal extracellular vesicles (AEVs) have not been systematically characterized, particularly for non-extremophilic archaea associated with the human host. Prior to this study, vesicle formation in these archaeal species had not been reported. This lack of data hinders our ability to differentiate archaeal contributions from bacterial ones within the gut ecosystem and to identify archaeal-specific mechanisms of host interaction.

Investigating AEVs could reveal novel signaling pathways and bioactive molecules unique to archaea, potentially uncovering distinct roles in microbiome stability and host modulation that are not observed with bacterial EVs.

In this manuscript, we focus on the herewith reported discovery of archaeal extracellular vesicles (AEVs) produced by human-associated archaeal representatives and present novel findings on their ultrastructure, proteome, and metabolome, as well as their interaction with human cell lines. We will discuss the implications of this discovery for our understanding of microbiome-host interactions and outline future directions for research.

## Results

### Archaeal extracellular vesicle (AEV) formation in all methanogen species

Vesicles produced by *Methanobrevibacter smithii* ALI, *M. intestini*, *M. smithii* GRAZ-2, and *Methanosphaera stadtmanae* were visualized by electron microscopy in culture supernatants in the late exponential/stationary phase. In detail, scanning-, negative staining- and ultra-thin electron-microscopy- based methods revealed the presence of vesicle-like structures within (Fig. 1g, j, k) and attached to the cells (Fig. 1a, e, i, l), as well as in their close vicinity (Fig. 1b, f) in all methanoarchaeal cultures. These were generally round shaped, approximately 87–198 nm in size (-130 nm on average, sizes measured during nanoparticle tracking analysis (NTA); Fig. 1c, d, h, m), and showed a clear, sharp edge. No vesicles were observed in culture media controls (MS medium), which underwent the same procedure for vesicle isolation including incubation and microscopy imaging (Supplementary Figs. S1 and S4a).

### Biophysical AEV characteristics

Using protocols developed and optimized for bacterial extracellular vesicle (BEV) analysis, AEV biomass production and isolation protocol (Supplementary Fig. S1) were established to enable characterization with respect to size, composition, ultrastructure, proteome, metabolome, and interaction with mammalian cells.

AEVs from the methanogens *M. smithii* ALI, *M. smithii* GRAZ-2, *M. intestini*, and *M. stadtmanae* were purified using a centrifugation, filtration, and concentration pipeline, previously established for bacterial BEVs[27] with minor adaptations (see materials and methods). To the former described BEV isolation protocol, a centrifugation step ($10,000 \times g$, 20 min) was added to remove residues from the culture media. The basic characteristics of the AEVs (size, concentration, nucleic acid, protein and lipid content) are summarized in Fig. 2 and Supplementary Data 1.

The size of the AEVs ranged from 86.9 to 197.3 nm with an average size of approximately 130 nm (Fig. 2a, Supplementary Data 1,2). *M. intestini*-derived AEVs were the largest (-136 nm on average), while those from *M. smithii* GRAZ-2 were the smallest (-117 nm), similar in size to vesicles from *M. smithii* ALI (-124 nm). Overall, the sizes of AEVs were within the general range reported for BEVs (20–400 nm), including vesicles from enterotoxigenic *Escherichia coli* (ETEC; 50–200 nm[29]) and *Bacteroides fragilis* (20–250 nm[30]), with measured averages of -120 nm and -194 nm, respectively (Supplementary Data 2). Statistical analysis revealed significant differences in AEV size between *M. smithii* ALI and *M. intestini* ($p = 0.046$), as well as *M. intestini* and *M. smithii* GRAZ-2 ($p = 0.007$; One-way ANOVA and Tukey's HSD *post hoc* test).

Average concentration of retrieved AEVs (Supplementary Data 1,2) was substantially lower ($1.4 \times 10^{10}$ to $3.7 \times 10^{11}$ particles/ml) than concentrations typically reported for BEVs, such as those measured for ETEC ($6.38 \times 10^{11}$ particles/ml), and *B. fragilis* ($8 \times 10^{11}$ particles/ml; Supplementary Data 3). Concentrations were reasonably consistent for both *M. smithii* strains, with lower concentrations retrieved for *M. intestini* and *M. stadtmanae* (Supplementary Data 1, 2). However, no statistically significant differences in AEV concentrations

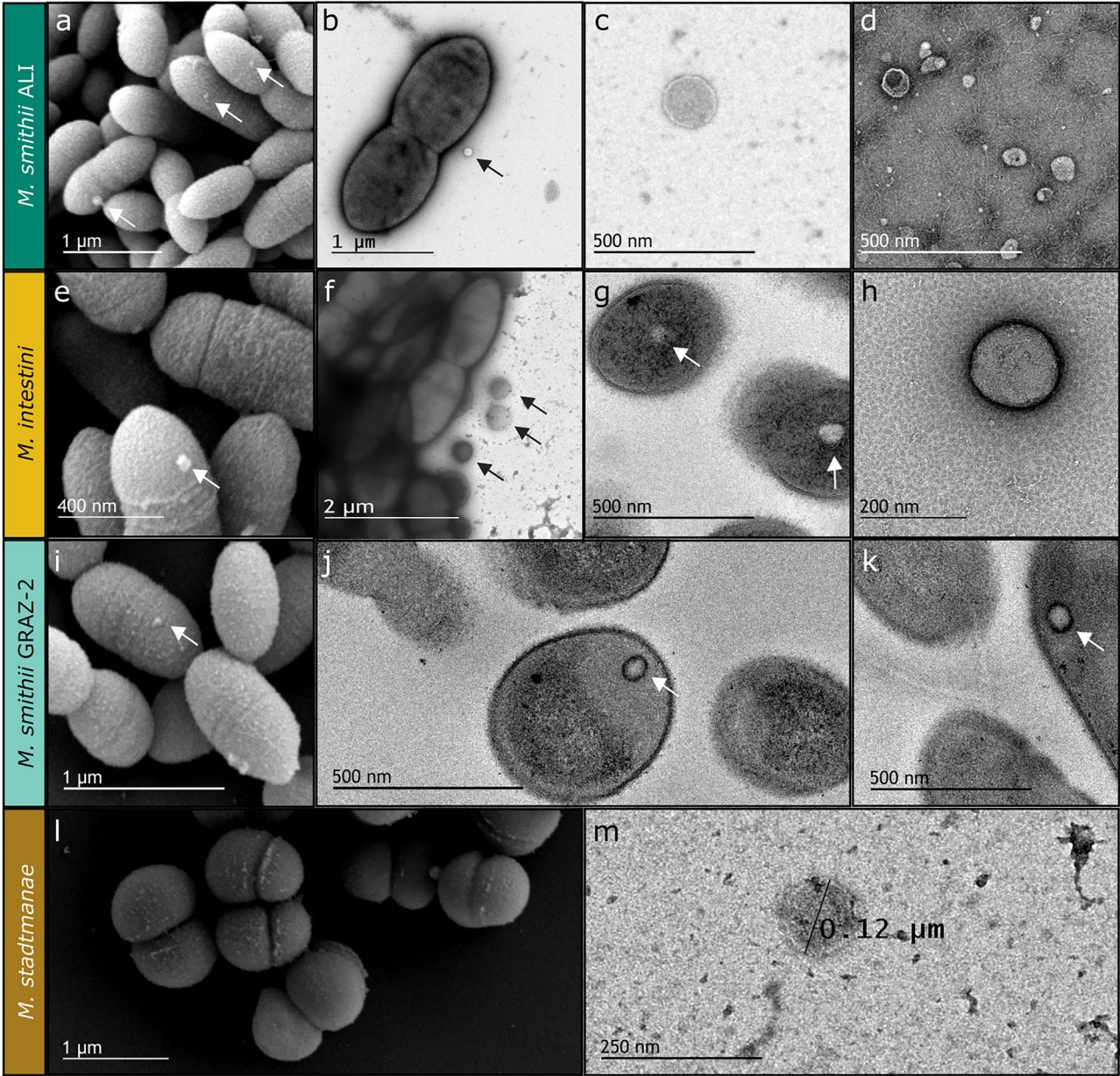

**Fig. 1 | Ultrastructure of archaeal cells and vesicle-like structures (*M. smithii ALI, M. intestini, M. smithii* GRAZ-2, *M. stadtmanae*). a**, **e**, **i**, **l** Scanning electron micrographs of whole cells showing potential vesicle development on their cell surface. **b**, **f**, **g**, **j**, **k** Ultra-thin transmission electron micrographs of whole cells, highlighting vesicles located inside or attached to the cells. **c**, **d**, **h**, **m** Transmission electron micrographs of isolated vesicles. Arrows indicate the presence of archaeal extracellular vesicles (AEVs). The experiments were repeated two times independently with similar results. Source data are provided as a Source Data file.

were found among the strains ($p > 0.05$, One-way ANOVA and Tukey's HSD *post hoc* test; Fig. 2b).

Protein content across AEVs ranged from 0.09 to 180.6 µg/$10^{10}$ particles (Supplementary Data 1, 2). On average, *M. smithii* GRAZ-2 vesicles contained the highest protein concentration (~52 µg/$10^{10}$ particles), while *M. smithii* ALI had the lowest (~7.8 µg/$10^{10}$ particles). Statistical analysis revealed no significant differences in the protein content of the strains ($p > 0.05$, One-way ANOVA, and Tukey's HSD *post hoc* test; Fig. 2c).

Overall, the DNA content of AEV extracts ranged from 0.004 to 18.27 ng/$10^{10}$ particles. *M. intestini* vesicles showing the highest DNA concentration (3.14 ng/$10^{10}$ particles), and *M. stadtmanae* the lowest (0.55 ng/$10^{10}$ particles) on average. Statistical analysis of DNA content showed no significant differences among the strains ($p > 0.05$, Kruskal–Wallis test, Fig. 2d). RNA was generally low and could not be

detected in all samples. *Methanobrevibacter*-derived AEVs contained similarly low amounts of RNA (0.08–0.014 ng/$10^{10}$ particles), while *M. stadtmanae* vesicles had slightly higher RNA levels (0.17 ng/$10^{10}$ particles) (Supplementary Data 1, 2). Statistical analysis revealed no significant differences in RNA content among the strains ($p > 0.05$, One-way ANOVA, and Tukey's HSD *post hoc* test, Fig. 2e).

Lipid content of AEVs was only partially within the range of the standard linoleic acid calibration (20–100 µg/ml). AEVs from *M. intestini* had the highest lipids content (~81.2 µg/$10^{10}$ particles), whereas *M. smithii* ALI vesicles contained the lowest (~4.9 µg/$10^{10}$ particles on average). As lipid content could only be measured in a limited number of samples (not all concentrations were within the standard range), no graphical display is shown, but all data are included in Supplementary Data 1,2. Statistical analysis of lipid content revealed no significant differences ($p > 0.05$, Kruskal–Wallis test).

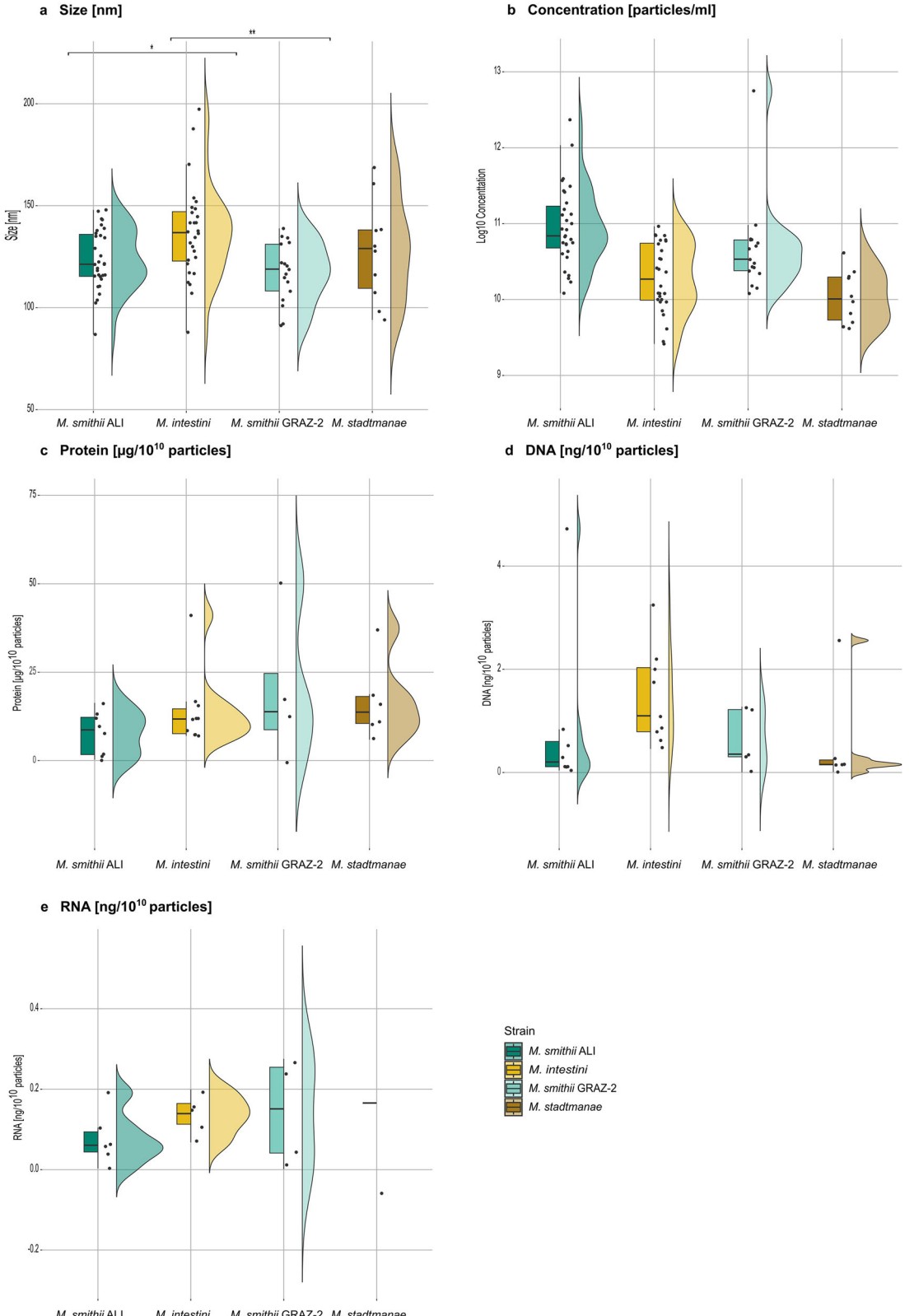

**Fig. 2 | Vesicle properties of *M. smithii* ALI, *M. intestini*, *M. smithii* GRAZ-2, and *M. stadtmanae*. a** Vesicle size [nm]: A one-way ANOVA followed by Tukey's HSD *post hoc* test revealed significant differences between *M. smithii* ALI and *M. intestini* ($p = 0.046$), as well as *M. intestini* and *M. smithii* GRAZ-2 ($p = 0.007$). No significant differences were observed for other comparisons. **b** Vesicle concentration [particles/ml]: No statistically significant differences were found between strains via one-way ANOVA and Tukey's HSD *post hoc* test ($p > 0.05$). Protein (**c**) was normalized to [µg/$10^{10}$ particles], and DNA (**d**) and RNA (**e**) content to [ng/$10^{10}$ particles] (Supplementary Data 3): A Kruskal-Wallis test was applied to DNA content (**d**), while one-way ANOVA and Tukey's HSD *post hoc* tests were performed for protein (**c**) and RNA (**e**) content. No significant differences were observed between strains for **c**, **d** or **e**. One outlier was removed for **d** and **e**. Dots represent the number of biological/technical replicates (Supplementary Data 2): *M. smithii* ALI, $n = 10/52$; *M. intestini* = 12/52; *M. smithii* GRAZ-2 = 5/26; *M. stadtmanae* = 6/17. The line inside the boxplot indicates the median, while the box spans the first to third quartile; whiskers represent the smallest and largest values within 1.5× the interquartile range. One asterisk represents $p < 0.05$, ** represents $p < 0.01$. Source data are provided as a Source Data file.

### *Methanobrevibacter* AEVs have comparable proteomes and show a massive enrichment in adhesins

Proteomic analysis was conducted exclusively on *M. smithii* ALI and *M. intestini*, as our focus was on the two predominant archaeal species in the human gut, thereby excluding *M. smithii* GRAZ-2 and *M. stadtmanae* from this specific analysis.

The protein cargo of *M. smithii* ALI and *M. intestini* AEVs were compared with their respective whole microbial cell proteomes (whole cell lysate, WCL). Profiling was carried out through LC-MS/MS, employing isolated AEVs and whole cell lysates (*n* = 3) of *M. smithii* ALI and *M. intestini*. A total of 1475 vesicular proteins across all isolated EVs was identified (*M. smithii* ALI: 801; *M. intestini*: 674), complemented by the identification of 2537 proteins from the whole cell lysates (WCL; *M. smithii* ALI: 1262; *M. intestini*: 1275, Supplementary Data 4, 5). Proteins were considered to be present in a sample, based on a prevalence in three out of three replicates for each group of samples (WCL *M. smithii* ALI: 1026; WCL *M. intestini*: 1100; EVs *M. smithii* ALI: 364, and EVs *M. intestini*: 259; Supplementary Fig. S2, Supplementary Data 5).

AEVs derived from *M. smithii* ALI (EV *M. smithii* ALI) and *M. intestini* (EV *M. intestini*) shared 229 proteins, while having 135 and 30 unique proteins, respectively (Supplementary Fig. S2b). Only a small number of proteins (*M. smithii* ALI: 35, *M. intestini*: 56) were exclusively detected in the vesicles but not in the whole cell lysates (Supplementary Fig. S2d, e). Proteins of whole cell lysates were highly similar, as 816 proteins were identified in both WCLs of *M. smithii* ALI and *M. intestini* (Supplementary Fig. S2a). 173 proteins were found in all four groups (EV and WCL of both species, Supplementary Fig. S2c). The principal component analysis (PCA) plot depicted in Fig. 3a illustrates different distribution patterns between whole-cell lysates and extracellular vesicles for both species. Notably, it also highlights the similarities observed between WCLs, as well as between EVs of *M. smithii* ALI and *M. intestini*.

The protein content of the vesicles of both *Methanobrevibacter* species was strikingly similar (Fig. 3, Supplementary Figs. S2 and S3), with 229 proteins that were prevalent in all six AEV samples, comprising three biological replicates each for *M. smithii* ALI and *M. intestini*. The most abundant proteins were adhesins/ adhesin-like proteins/ proteins (ALPs) with an IG-like domain, as identified through InterPro prediction (Fig. 3b, c; Supplementary Data 7, 8)[31]. These proteins were also highly enriched compared to the whole-cell lysates (WCL, Fig. 3b, c; Supplementary Data 6)[31].

The proteomic profile of the archaeal cell membrane fraction (MF) was analyzed to investigate a potential enrichment in ALPs. Proteins were considered to be present based on a prevalence in at least 4 out of 5 replicates per species (*M. smithii* ALI, *M. intestini*). Overall, approximately 4% of the proteins identified in the cell membrane fractions of *M. smithii* ALI and *M. intestini* were annotated as ALPs (total identified proteins = 1904, number of ALPs = 81; Supplementary Data 8). In contrast, ALPs accounted for 20% of the identified proteins in vesicles (total identified proteins = 229; ALPs = 46; Supplementary Data 6). Among the 46 ALPs identified in AEVs, 23 for *M. smithii* ALI and 41 for *M. intestini* were also found in their respective cell membranes (Supplementary Data 9). Additionally, the ALP profiles of *M. smithii* ALI and *M. intestini* shared 37 ALPs, while approximately 37% of their ALP profiles were distinct, highlighting species-specific differences (total ALPs *M. smithii* ALI: 57; total ALPs *M. intestini*: 60; Supplementary Data 9).

ALPs are rarely studied in archaea, but were found to be very abundant in e.g. rumen methanogens where they account for up to 5% of all genes[32]. It has been suggested that the ALPs serve to attach to their protozoan hosts or to the cell surface of bacteria[33]. ALPs have also been found in human-associated *Methanobrevibacter* species[34], for which adhesion and sugar-binding function has been proposed. Indeed, the identified vesicle-associated ALPs carried a variety of protein motifs, indicative of adhesive (invasin/intimin cell-adhesion fragments; IG-like_fold superfamily) and polysaccharide binding functions (PbH1;

pectin_lyase_fold, Pectin_lyase_fold/virulence; details for all genes and their identified motifs are given in Supplementary Data 7).

Bacterial proteins containing IG-like domains exhibit a broad spectrum of functions, such as cell host adhesion and invasion. IG-like domains are also found in periplasmic chaperones and proteins that assemble fimbriae, in oxidoreductases and hydrolytic enzymes, ATP-binding cassette transporters, sugar-binding and metal-resistant proteins[35]. These proteins are structural components of bacterial pilus and nonpilus fimbrial systems and members of the intimin/invasin family of outer membrane adhesins, indicating their relevance for adhesion and interaction with the biological surroundings[35]. Microbial pectin and pectate lyases are involved in the degradation of pectic components of the plant cell, which is an important trait for plant pathogens, as well as the degradation of dietary components in the GIT[36]. However, this specific β-helix topology has various functions e.g. as galacturonases, or for the adhesion to mammalian cells[36].

Within a group of transport-associated proteins, we found substantial enrichment of a protein (representative: GUT_GENOME043902_01504) with an oligopeptide transporter (OPT) superfamily domain (Fig. 3b; Supplementary Data 6, Supplementary Fig. S5). In general, OPT transporters are known for oligopeptide uptake but can also facilitate the transport of iron-siderophore complexes[37], indicating a potential role in iron uptake.

A further substantial increase was observed for a putative DUF11 domain-containing protein[38] (Fig. 3b, Supplementary Data 6, Supplementary Fig. S5), which might be important for stabilizing surface wall structures in *Methanothermobacter* sp. strain CaT2[38]. Another interesting finding was the increased presence of a putative peptidase_C1 (Fig. 3b; Supplementary Data 6, Supplementary Fig. S5), which also showed adhesin-like domains (Supplementary Data 7).

### Metabolite cargo of AEVs

Similar to the proteomic analyses, the metabolic profiles of AEVs of *M. smithii* ALI and *M. intestini* were overall similar, but with high variability across biological replicates, presumably due to variations in input concentrations (see group coefficient of variation (CV) % in Supplementary Data 10; Fig. 4). Strikingly, the AEVs of *M. intestini* revealed a significantly increased content of free glutamic and aspartic acid (Fig. 4; *p* = 0.03 and *p* = 0.01, respectively; Supplementary Data 10; this table also includes details on statistics). Similarly, the AEVs of *M. smithii* ALI revealed a noticeable but statistically non-significant increase of these amino acids compared to the negative control (incubated, empty medium) (Fig. 4). Further, the AEVs of *M. smithii* ALI were substantially loaded with arginine (Fig. 4).

Notably, glutamic acid has been identified as a component of BEVs (*B. fragilis*)[39]. Besides their roles in central metabolism, both amino acids (glutamic and aspartic acid) are considered to act as neurotransmitters[40]. Glutamic acid plays a fundamental role as an excitatory neurotransmitter in the central and enteric nervous system and acts, together with other metabolites, along the microbiota-gut-brain axis[41] as an interkingdom communication system. It is considered that the glutamatergic receptors, along the microbiota-gut-brain axis, could have an impact on multiple physiological responses in the brain and gut. As glutamic acid usually does not enter the bloodstream from the large intestine, AEVs could be supporting the transmission to glutamatergic enteric neurons/receptors[41]. Despite its potential function as a neurotransmitter, aspartate also supports the proliferation of mammalian cells (e.g. cancer cells)[42].

Choline glycerophosphate (glycerophosphorylcholine, alpha GPC) was found to be elevated in AEVs of both species (Fig. 4). Also, for this compound, a potential neurological effect was described, which has been considered for the treatment of Alzheimer's disease[43].

The origin of the salicylic acid, which was found to be increased in AEVs of both species, is unclear (potentially derived from chorismate), but its effects on the host and microbiome could include bactericidal

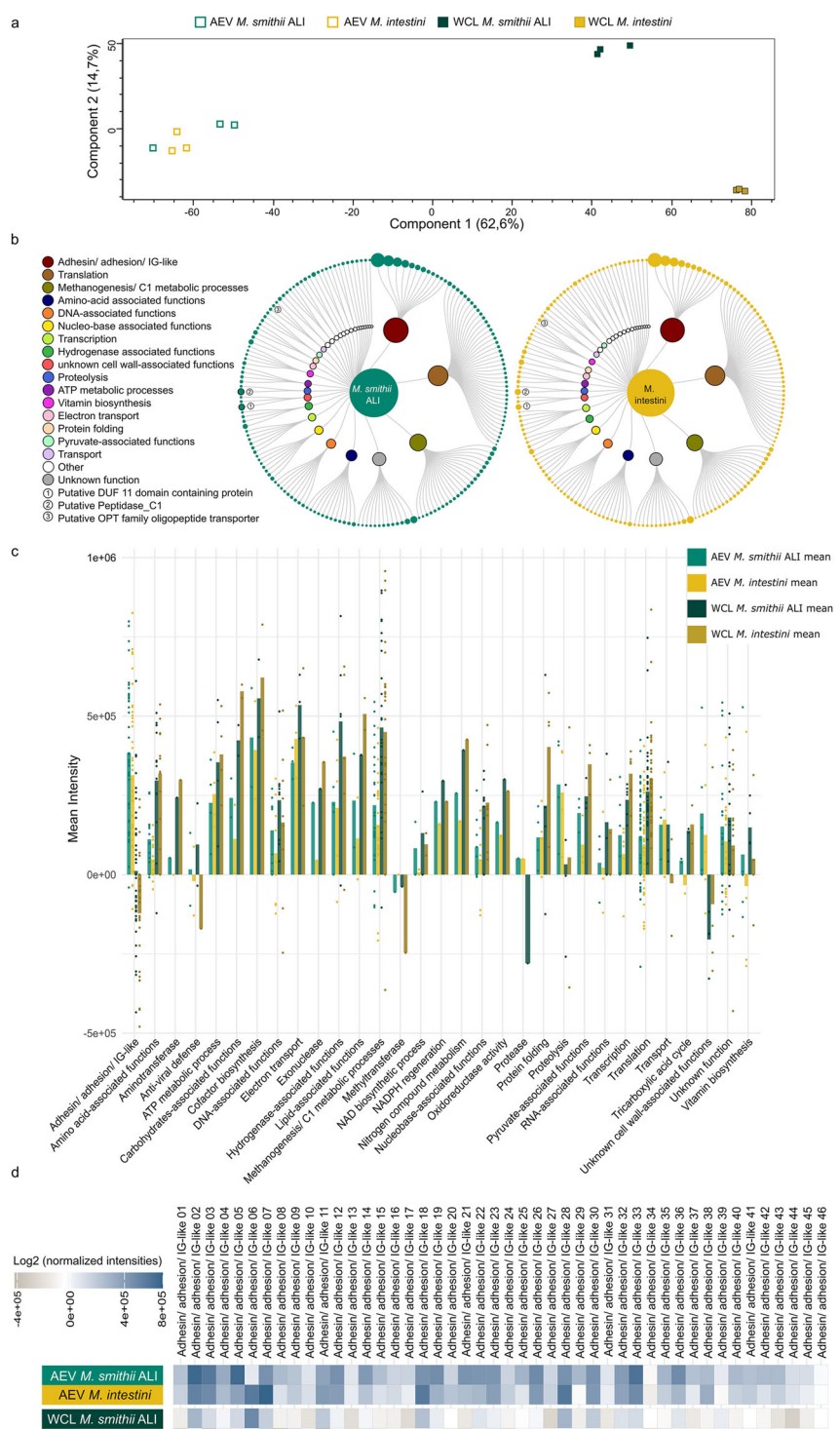

**Fig. 3 | Mass spectrometry-based profiling of AEV proteomes. a** Principal component analysis (PCA) plot illustrating the protein profiles of AEVs and WCLs of *M. smithii* ALI and *M. intestini*, including only proteins detected in all three replicates per group (AEV *M. smithii* ALI, AEV *M. intestini*, WCL *M. smithii* ALI, WCL *M. intestini*). **b** Overlap of 229 proteins identified in the AEVs of *M. smithii* ALI (left, *n* = 3 biological replicates) and *M. intestini* (right, *n* = 3 biological replicates), visualized and organized by intensities/relative abundance (circle size) and functional categorization (see Supplementary Data 6–8 for details). Data were visualized using

RawGraphs[113] and InkScape[104]. **c** Bar chart displaying mean intensities/relative abundances of protein categories in AEVs and WCLs, based on proteins detected in all three biological replicates of both *M. smithii* ALI and *M. intestini* (*n* = 229) (Supplementary Data 6–8). **d** Heatmap showing enrichment of 46 proteins annotated as adhesin/adhesion/IG-like present in all six AEV extracts (three biological replicates each of AEV *M. smithii* ALI and AEV *M. intestini*) compared to the whole cell lysates based on relative abundances. WCL whole cell lysate, AEV archaeal extracellular vesicles. Source data are provided as a Source Data file.

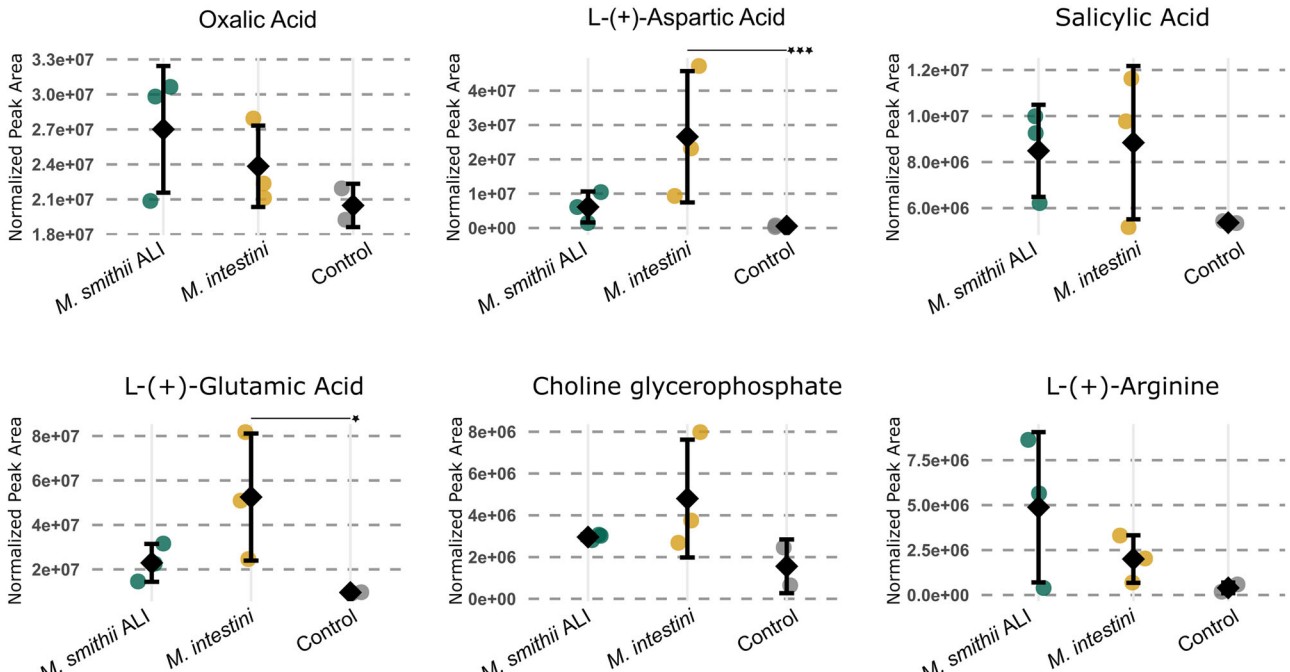

**Fig. 4 | Metabolites detected in archaeal vesicles.** Metabolite levels in AEVs from three biological replicates per species, compared to a non-cultured control medium processed through the vesicle isolation pipeline (technical duplicates) (Supplementary Data 10). The Y-axis represents the normalized peak area (LC-MS). Significantly changed compounds are marked with an asterisk (aspartic acid:

$p = 0.0010$ (***); glutamic acid: $p = 0.0276$ (*); Tukey HSC *post hoc* test after ANOVA, which applies a correction for multiple comparisons. All tests were two-sided). Data points represent individual measurements; black diamonds indicate the group means, and error bars show standard deviation. Source data are provided as a Source Data file.

and antiseptic action in higher concentrations[44]. Another compound found to be increased was oxalic acid, having the characteristics of a chelating agent for metal cations, making insoluble iron compounds into a soluble complex ion, which could be an interesting trait for gastrointestinal microbiota[45].

## Human macrophages acquire AEVs

Human leukemia monocytic THP-1 cells are a common model for studying monocyte/macrophage functions, signaling pathways, mechanisms, and drug and nutrient transport[46]. For visualizing the association or interaction of AEVs with host cells, AEVs of *M. smithii* ALI, *M. intestini*, *M. smithii* GRAZ-2, and *M. stadtmanae* were incubated with macrophage monolayers (differentiated THP-1 cells) for 24 h and their localization was assessed by immunofluorescence microscopy. Co-localization of DiO-labeled AEVs (Fig. 5a–d, green dye to label the AEVs membrane) with host cell nuclei were investigated using the nuclei marker Hoechst 33342 (blue), and the cytoskeleton marker Alexa 647-Phalloidin (red). AEVs from all strains were shown to be in close association with the nuclei (Fig. 5a–d). Similar localization of EVs was previously described for BEVs e.g. from *B. thetaiotaomicron*[47]. A representative z-stack of *M. stadtmanae* AEVs and macrophage monolayers supports the uptake of AEVs by the macrophage cells (Fig. 5e). Here, AEVs were additionally labeled with specific polyclonal anti-archaea antibodies and an Alexa Fluor 647 (AF647)-labeled secondary antibody (red). Hoechst 33342 (blue) was used as a nuclei marker. Co-localization of the archaeal antibodies and the DiO is visible in yellow (Fig. 5e).

## AEVs induce various chemokines and cytokines in macrophages and epithelial cells

To investigate the immunostimulatory potential of AEVs, we examined inflammatory responses in immune cells (differentiated THP-1) and epithelial cells (HT-29) upon exposure to AEVs derived from *M. smithii* ALI, *M. smithii* GRAZ-2, *M. intestini*, and *M. stadtmanae*. A total of 23 inflammatory cytokines and chemokines were chosen as they

demonstrated a robust BEV-dependent induction in a previous study[25]. Moreover, BEVs of ETEC and *B. fragilis* were included as representatives of intestinal BEVs known to induce a high proinflammatory (ETEC) and a reducing anti-inflammatory (*B. fragilis*) response[48–50]. Two EV doses ($10^8$ and $10^9$ particles/ml cell culture medium) were tested for the induction of an immune response in host cells (Fig. 6, Supplementary Data 11, 12).

In THP-1 macrophages, we observed a species-dependent induction of inflammatory chemokines (Fig. 6, Supplementary Data 11, 12). For instance, chemokines involved in immune cell migration, activation, and the regulation of immune responses[51] (such as CXCL9 and CXCL11) were specifically triggered by various AEVs. CXCL9 was highly induced by AEVs derived from *M. smithii* GRAZ-2 and *M. stadtmanae*, whereas CXCL11 was strongly induced by AEVs from both *M. smithii* strains. Notably, AEVs from *M. intestini* and *M. stadtmanae* even led to a reduction of CXCL11 excretion compared to the respective control. Similar effects could be detected for AEVs from *M. smithii* ALI and *M. intestini* in case of CXCL9 response.

For pro-inflammatory cytokines, like TNF-α and IL-6, involved in various inflammatory processes and conditions, distinct patterns were observed. All AEVs, except those from *M. smithii* ALI, led to elevated levels of TNF-α. In contrast, AEVs from *M. smithii* ALI, *M. intestini*, and *M. stadtmanae* induced comparatively lower levels of IL-6, while AEVs from *M. smithii* GRAZ-2 elicited higher IL-6 levels. Additionally, a dose-dependent TGF-β response was observed for AEVs derived from *M. intestini* and *M. smithii* GRAZ-2. TGF-β is an important cytokine with diverse roles in cellular regulation and immune modulation; in cancer it shows a dual role as both, as suppressor and promotor[52].

We also tested the ability of AEVs to induce an inflammatory response on human intestinal HT-29 cells, derived from human colorectal adenocarcinoma. These epithelial cells serve as a valuable model for studying intestinal epithelial cell behavior and inflammatory responses. Widely used in gut-related research, HT-29 cells are well-suited for investigating interactions with extracellular vesicles and

their impact on inflammation and have recently been used in a comparative study to assess the differential pro-inflammatory potency of BEVs derived from gut bacteria[48,53]. In concordance with a recent report[54], AEVs derived from *M. smithii* ALI, *M. smithii* GRAZ-2, and *M. stadtmanae* failed to induce a notable increase of IL-8 levels, AEVs from *M. smithii* GRAZ-2 showed even negative/inhibitory effects. In contrast,

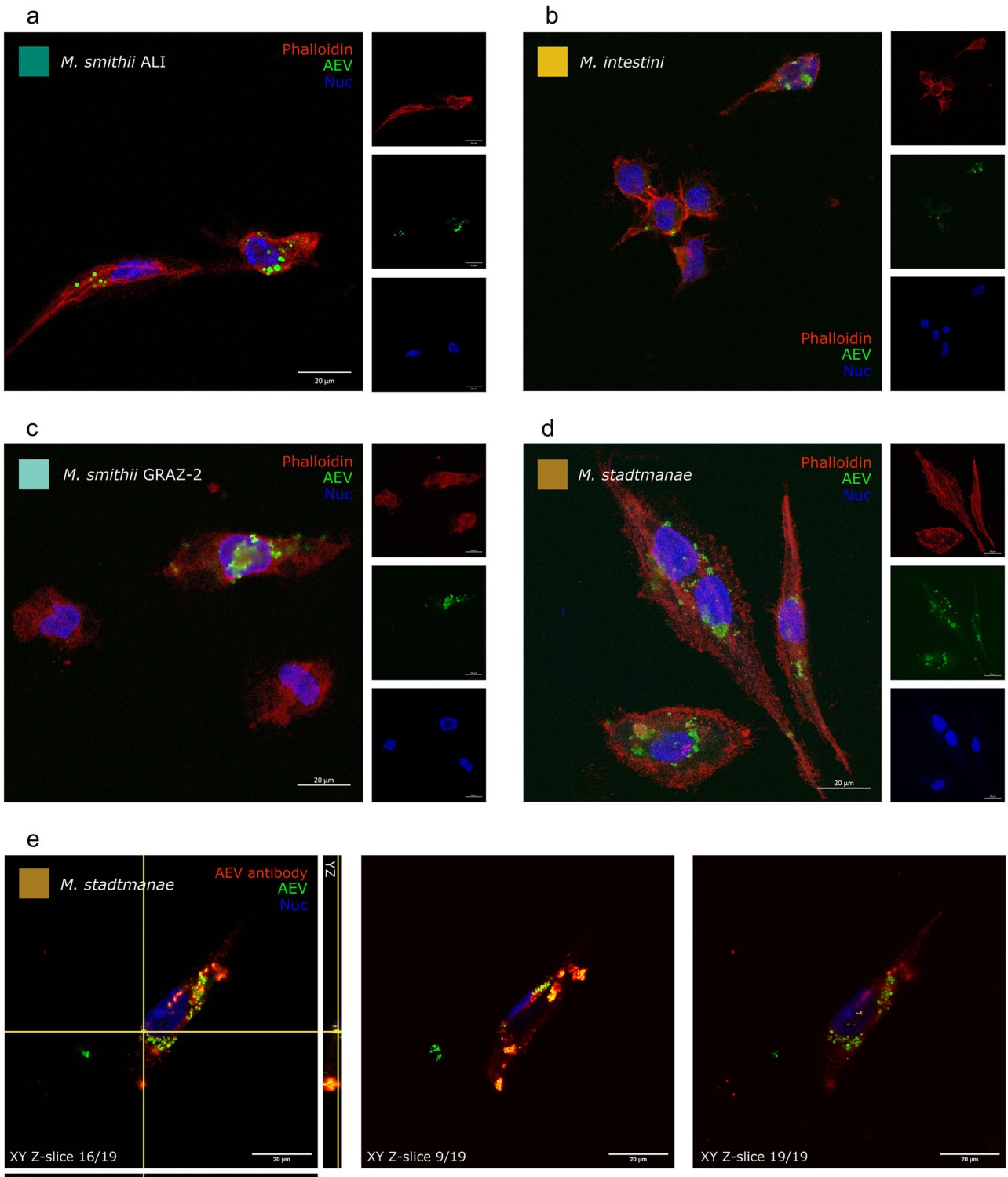

**Fig. 5 | Immunofluorescence microscopy of DiO-labeled AEV uptake by human macrophages.** Human macrophages incubated for 25 h with DiO-labeled (green) AEVs derived from **a** *M. smithii* ALI, **b** *M. intestini*, **c** *M. smithii* GRAZ-2, and **d** *M. stadtmanae*. Macrophage monolayers were stained with antibodies to visualize cytoskeleton (Alexa 647-Phalloidin, red), nuclei (Hoechst 33342, blue).
**e** Representative z-stack of *M. stadtmanae* AEVs internalized by a macrophage. AEVs were additionally labeled with specific anti-archaea antibodies (red) and macrophages were stained with Hoechst 33342 (blue) for visualization of the nuclei. Orthogonal views (XY, YZ and X-Z; left panel) show the plane of view (yellow lines) with example z-slice images at different z-depths (right panel). Images were acquired using a Zeiss LSM880 confocal microscope equipped with a 63 × /1.40 oil objective. Scale bar 20 μm. Experiments were performed twice for each strain. Source data are provided as a Source Data file.

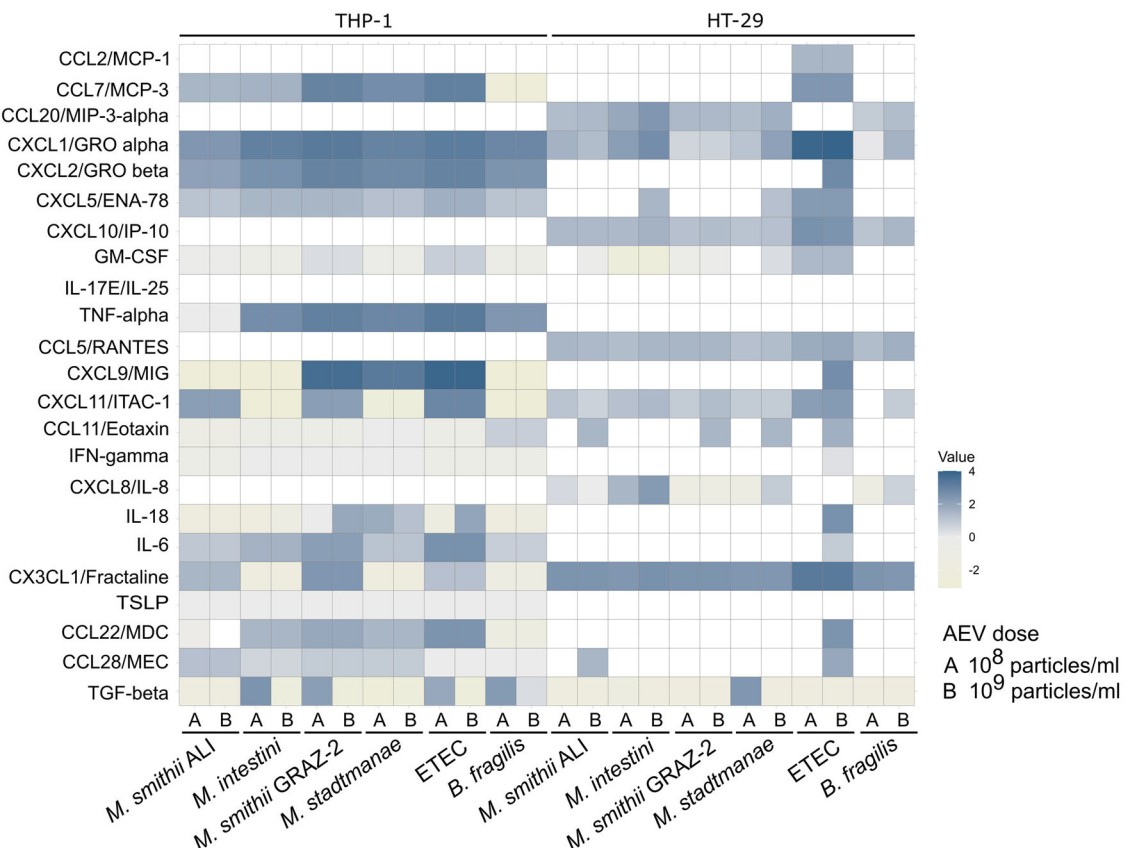

**Fig. 6 | Heat map showing the induction of cytokine release by macrophages (differentiated THP-1 cells), and intestinal epithelial cells (HT-29).** Cell lines were exposed to archaeal (*M. smithii* ALI, *M. intestini*, *M. smithii* GRAZ-2, and *M. stadtmanae*) and bacterial (ETEC, *B. fragilis*) EVs. Cytokine levels were measured by Luminex analyses from supernatants of HT-29 and THP-1 cells exposed to the different EVs for 24 h. Individual cytokines are indicated on the left. EV dose is indicated (**A**) $10^8$ particles/ml cell culture medium and **B** $10^9$ particles/ml cell culture medium on the bottom, as well as the vesicle origins. Saline (no EVs) for HT-29 and DMSO for THP-1 cells served as no treatment controls (NTC) to determine non-stimulated secretion levels of cytokines in the respective cell line, which were subtracted as blank from samples. Display of values is log10 transformed. Blue boxes show high levels, and yellow/beige boxes show negative levels of cytokine release upon EV exposure (see scale). White boxes indicate no measurable induction of cytokine release (Supplementary Data 12). Source data are provided as a Source Data file.

exposure of HT-29 cells to AEVs derived from *M. intestini* resulted in a substantial IL-8 induction for both doses. IL-8 is a chemokine that plays a crucial role in attracting neutrophils and other immune cells to sites of infection or inflammation[55]. Moreover, all AEVs resulted in a relatively high CX3CL1 response. These results suggest that AEVs derived from different archaeal species demonstrate a distinct inflammatory potency in HT-29 cells. Notably, the highest chemokine response, CX3CL1, was observed in HT-29 cells after exposure to AEVs from all four archaeal strains (*M. smithii* ALI, *M. intestini*, *M. smithii* GRAZ-2, and *M. stadtmanae*).

## Discussion

The discovery of archaeal extracellular vesicles (AEVs) produced by human gastrointestinal tract (GIT)-associated archaea introduces a novel principle in archaea-microbiota and archaea-host interactions. Similar to bacterial extracellular vesicles (BEVs) from the human gut microbiome, AEVs are membrane-bound structures that transport various biomolecules, including proteins, lipids, and nucleic acids[4,8–12]. We propose that these vesicles play a role in modulating microbial communities and host physiology by serving as a communication and cargo vehicle.

Both studied archaeal genera, *Methanobrevibacter* and *Methanosphaera*, were capable of vesicle formation. All vesicles were readily acquired by human macrophages (Fig. 5), and stimulated the secretion of various cytokines and chemokines in both macrophages and

intestinal epithelial cells (Fig. 6). For detailed proteomic and metabolomic studies, this study focused on vesicles from *Methanobrevibacter* species, the most abundant archaea in the human gut microbiome, comprising up to 4% of the microbiome[56]. *Methanobrevibacter* species rely on syntrophic bacterial partners that provide small molecules like $H_2$ (or formate) and $CO_2$ for methanogenesis[34,57,58]. The bacterial partner benefits from this interaction, as potentially inhibiting end products of fermentation are efficiently removed[34,57,58]. As such, a well-regulated and controlled interaction with bacterial syntrophic partners is highly crucial for *Methanobrevibacter* species.

This study demonstrates that AEVs derived from GIT-associated archaea are comparable in size to BEVs, although the particle count was substantially lower for archaea (Supplementary Data 2). Previous research has indicated that growth conditions, such as growth stage and medium composition, can influence the particle count, size, and vesicle cargo of BEVs leading to a heterogeneity among BEVs[13,59–65]. It is likely that similar effects occur with AEVs. Importantly, adhesins/ adhesin-like proteins (ALPs) were highly enriched in archaeal vesicles. These have been proposed to be important communication molecules in microbial interactions. For instance, *Methanobrevibacter* influences the metabolism of *Christensenella minuta*, shifting short-chain fatty acid (SCFA) production from butyrate to acetate[66]. This complex communication system, regulating the metabolic processes of both partners, is believed to be mediated by *Methanobrevibacter* surface adhesins, leading to significant physiological changes in the involved

microorganisms[66]. From the bacterial domain, numerous adhesins are known to mediate interaction, colonization, infection and host interaction, making them key targets in bacterial pathogenesis[67,68]. Considering that adhesins/adhesin-like proteins are highly enriched in AEVs, as shown in Fig. 3b–d, an important role of AEVs for archaeal-bacterial and archaeal-host interactions over longer distances is likely.

In *Methanobrevibacter ruminantium*, a prevalent *Methanobrevibacter* species in ruminants, 5% of its genome is predicted to encode putative adhesins or adhesin-like proteins[32]. Among these, Mru_1499 is essential for binding and interacting with hydrogen-producing protozoa and bacteria, such as *Butyrivibrio proteoclasticus*, to enhance methane production[33]. Additionally, adhesins were found to be upregulated not only during syntrophic interactions with hydrogen-producing microorganisms, but also under nicotinic acid (vitamin B3) limitation[32,33,69], indicating a complex interplay between metabolite availability and microbial or host interactions. Enriching adhesins on mobile vehicles such as AEVs offers numerous benefits, including the ability to reach communication partners beyond the immediate physical proximity of the non-motile archaeal cells potentially enabling even a global regulation of bacterial metabolism.

The interaction of AEVs with host cells is another key finding, as evidenced by their efficient uptake in human monocytes and the cytokine/chemokine observed in immune and epithelial cells (Figs. 5 and 6). However, further studies are needed to explore the uptake in other cell types, such as epithelial cells, and to assess the impact of different AEV types and compositions on the mechanisms of entry into host cells. Given that archaeal adhesins are believed to be heavily glycosylated[70], species-specific glycosylation patterns may explain the partially differing responses of HT-29 cells to AEVs from *M. smithii* ALI and *M. intestini*, despite similar overall AEV composition (Figs. 2 and 3). These findings highlight the need for further research into adhesin glycosylation patterns and their implications for host-microbe interactions.

The metabolic profiling of AEVs revealed increased levels of aspartic and glutamic acid (Fig. 4), probably suggesting a potential link between AEVs and the gut-brain axis (as discussed in the results section). The possibility that AEVs could influence host neurological processes warrants further investigation. Given that BEVs are known to interact with their neighboring cells, cross the intestinal barrier, and enter the bloodstream, potentially reaching distant tissues such as the brain[10,71], it is plausible that AEVs exhibit similar properties.

These characteristics make AEVs promising candidates for future applications such as drug delivery vehicles or targeted therapeutic systems, as demonstrated previously with EVs of probiotic bacteria[72–74]. Moreover, similar to BEVs, AEVs could serve as postbiotics, microbial-derived substances that confer beneficial effects on the host[75].

While this study provides substantial insights into the role of AEVs in microbial communication and host interactions, several limitations must be acknowledged. The DNA and RNA content of vesicles was measured without prior DNase or RNase treatment, meaning the results may include measurements of surface-attached nucleic acids. Further, the analyzed AEVs were derived from monocultures grown in artificial growth media, which may not fully represent their natural state in the GIT. As a result, findings from e.g. metabolomics and proteomics should be validated using host-isolated vesicles, though archaeal vesicles are likely underrepresented in such samples. Additionally, while biological replicates were used, natural fluctuations in vesicle cargo composition were observed, likely due to vesicle heterogeneity. Different vesicle subtypes may carry distinct cargo, leading to varied biological effects, depending on the targeted microbial or host cells[76–78]. Moreover, the isolation process itself might impact the retrieval of different vesicle subtypes[76].

## Methods

### Source of microorganisms
The human gut derived strains *Methanobrevibacter smithii* ALI (DSM 2375) and *Methanosphaera stadtmanae* (DSM 3091, type strain) were obtained from the German Collection of Microorganisms and Cell Cultures (DSMZ) GmbH, Braunschweig, Germany. *M. intestini* WWM1085 (DSM 116060) was obtained from the Department of Microbiology, University of Illinois, USA, where it was isolated from a stool sample[79]. *M. smithii* GRAZ-2 (DSM 116045) was isolated in 2018 at the Medical University of Graz, Graz, Austria, from a stool sample of a healthy woman[79]. Instead of opting for the *Methanobrevibacter smithii* type strain (PS, DSM 861), our choice was *M. smithii* ALI, as it sourced from a human fecal sample and not from sewage water. Enterotoxigenic *Escherichia coli* (ETEC) H10407 and *Bacteroides fragilis* ATCC® 25285 have been reported previously[48].

### Growth media and cultivation
For the cultivation of all methanogens standard MS medium was used with some modifications as previously described[79]. For vesicle production, aliquots of 250 ml media in 1000 ml infusion bottles were sealed, pressurized with $H_2/CO_2$ (4:1) and autoclaved. Before inoculation and incubation at 37 °C, sodium acetate (0.001 g/ml, anoxic, sterile) and yeast extract (YE, 0.001 g/ml, anoxic, sterile) were added to the media.

### Electron microscopy
Electron microscopy (EM) was conducted at the Core Facility Ultrastructure Analysis, Medical University of Graz, Graz, Austria and at the Core Science Resources Quadram Institute Bioscience, Norwich, United Kingdom. For ultrastructural analyses of cells, isolates were cultivated in 20 ml aliquots in 100 ml serum bottles for 7 days under anaerobic conditions at 37 °C in an incubation shaker (shaking speed: 80 rpm). Next, 2 ml of medium containing each strain was centrifuged at $4000 \times g$, 4 °C, for 10 min. Cell pellets were then directly handed over to the Core Facility for further preparation. AEVs ($1 \times 10^{11}$/ml) were directly handed over to the Core Science Resources Quadram Institute Bioscience, Norwich, United Kingdom.

### Transmission electron microscopy: thin sections and tomography
Cells were fixed in 2.5% (w/v) glutaraldehyde and 2% (w/v) paraformaldehyde in 0.1 M cacodylate buffer, pH 7.4, for 1 h, postfixed in 1% (w/v) osmium tetroxide for 2 h at room temperature, dehydrated in graded series of ethanol and embedded in TAAB (Agar Scientific) epoxy resin. Ultrathin sections (70 nm thick) were cut with a UC 7 Ultramicrotome (Leica Microsystems) and stained with lead citrate for 5 min and with platinum blue for 15 min. Images were taken using a Tecnai G2 20 transmission electron microscope (Thermo Fisher Scientific) with a Gatan ultrascan 1000 charge coupled device (CCD) camera (temperature −20 °C; acquisition software Digital Micrograph; Gatan). The acceleration voltage was 120 kV. The tilt series was reconstructed using FLARA, a joint alignment and reconstruction algorithm for electron tomography. This iterative algorithm allows for acquisitions without fiducial gold markers, since an effective shift computation can be obtained by using a global alignment technique based on a linearized approximation of the disruptive shifts in each iteration[80]. For negative staining cell suspensions were placed on glow discharged carbon coated copper grids for 1 min. The solution was removed after incubation by filter paper stripes. A drop of 1% aqueous uranyl acetate solution was placed afterwards for 1 min, dried with filter paper and later on air dried at room temperature. Specimens were examined with an FEI Tecnai G 2 (FEI) equipped with a Gatan ultrascan 1000 charge coupled device (CCD) camera (−20 °C, acquisition software Digital Micrograph, Gatan).

AEV suspensions were visualized using negative staining with TEM. Briefly, 4 µl AEV suspension was adsorbed to plasma-pretreated carbon-coated copper EM grids (EM Solutions) for 1 min before wicking off with filter paper and negatively staining with 1% Uranyl Acetate solution (BDH 10288) for 1 min. Grids were air-dried before analysis using a FEI Talos F200C electron microscope at 36,000 × −92,000x magnification with a Gatan OneView digital camera.

## Scanning electron microscopy

For scanning electron microscopy, cells were affixed to coverslips and treated with a fixing solution consisting of 2% paraformaldehyde and 2.5% glutaraldehyde in 0.1 M phosphate buffered saline (pH 7.4). Subsequently, a graded ethanol series was used for dehydration. Post-fixation involved 1% osmium tetroxide for 1 hour at room temperature, followed by additional dehydration in an ethanol series (ranging from 30% to 100% EtOH). Hexamethyldisilazane (HMDS) was applied, and coverslips were positioned on stubs using conductive double-coated carbon tape. Imaging was performed with a Sigma 500VP FE-SEM equipped with a SE Detector (Zeiss Oberkochen) operating at an acceleration voltage of 5 kV.

## Archaeal extracellular vesicle (AEV) Isolation

To obtain a sufficient amount of biomass for the isolation of AEVs, 250 ml of MS medium was aliquoted into 1000 ml infusion bottles (VWR) and further handled the same way as described above. These cultures were then cultivated for 10 days under anaerobic conditions at 37 °C in an incubation shaker (shaking speed: 80 rpm). When the pressure of cultivation bottles dropped due to growth, they were re-gassed with $H_2/CO_2$. Growth was surveyed by optical density photometry at 600 nm. On day ten, the cell suspensions were centrifuged at 14,000 × g, 4 °C, 20 min (Thermo Scientific™ Sorvall™ LYNX™ 6000). To remove cell debris and remaining cells, the supernatant was filtered with 0.22 µm PES bottle-top filters (Fisherbrand™ Disposable PES Bottle Top Filters). If not immediately processed, the supernatant containing the vesicles was stored at 4 °C overnight. For long-term storage, vesicles were stored at −20 °C. Vesicles were freshly prepared for each experiment.

Isolation of vesicles was done according to Stentz et al.[81] (work-flow see Supplementary Fig. S1). In brief, a filtration cassette (Vivaflow 50R, 100,000 MWCO, Hydrostat, model VF05H4, Sartorius or Vivaflow 200 100,00 MWCO, PES, model VF20P4, Sartorius) was used to concentrate 1 L of sample down to approx. 5 ml. Then, 500 ml PBS buffer (pH 7.4) was added for washing purposes, and the liquid was concentrated to 1-4 ml. The sample was then centrifuged for 20 min at 10,000 × g, 4 °C to remove protein and lipid aggregates. Next, the sample was transferred to Pierce™ Protein Concentrators (PES, 100,000 MWCO, Thermo Scientific) and centrifuged at 3000 × g, 4 °C until the samples were concentrated down to 1 ml. Residual contaminants and proteins were further eliminated through size exclusion chromatography (SEC) using an IZON qEV1 column (pore size 35 mm) according to the manufacturer's instructions. The vesicles were eluted in the 2.8 ml fraction containing the purified extracellular vesicles underwent a final filter sterilization using a 0.22 µm syringe filter (ROTILABO® PES), and were subsequently stored at 4 °C until further use.

To ensure that the final AEV suspension does not contain any yeast vesicles or other residues, the YE was subjected to cross-flow ultrafiltration (100 kDa MWCO).

For the metabolomics analyses, 1 L of blank MS medium underwent the same procedure to serve as a control.

## Bacterial extracellular vesicle (BEV) Isolation

BEVs from ETEC and *B. fragilis* for the HT-29 experiment were isolated as described previously with minor modifications[48,82]. Briefly, overnight cultures were either grown with aeration (180 rpm, Infor shaker)

in case of ETEC or anaerobically (GasPak™ EZ Systems, BD) in case of *B. fragilis* to ensure sufficient growth. The respective cultures were diluted (1:100) in BHI medium and grown at 37 °C either with aeration for 8 h or overnight anaerobically (GasPak™ EZ Systems, BD). The cells were then removed from the supernatant by centrifugation (9000 × g, 15 min, RT) and subsequent sterile filtration (0.22 µm). The BEVs present in the supernatant were pelleted through subsequent ultracentrifugation (150,000 × g, 4 °C, 4 h), resuspended in appropriate volumes of PBS to generate a BEV suspension 1000-fold more concentrated than in the original culture supernatant. Quantification and size distribution of BEVs were investigated by nanoparticle tracking analysis (NTA) using a Nanosight NS300 (see below).

## AEV characterization

**Nanoparticle tracking analysis (NTA).** Quantification and size distribution of AEVs were investigated by nanoparticle tracking analysis (NTA) using ZetaView and Nanosight NS300. ZetaView was used by following established protocols[59,81]. In brief, particles were quantified using the ZetaView instrument (Particle Metrix) with ZetaView (version 8.05.12 SP1) software running a two cycle 11 position high frame rate analysis at 25 °C. Samples were diluted with ultrapure water allowing the optimal detection range. Camera control settings: 80 Sensitivity; 30 Frame Rate; 100 Shutter. Post-acquisition parameters: 20 Min Brightness; 2000 Max Area; 5 Min Area; 30 Trace Length; 5 nm/Class; 64 Classes/Decade.

For NanoSight NS300 (Malvern Instruments) samples were diluted in 1x PBS according to the manufacturer's guidelines (final concentration between $10^7$–$10^9$ particles per ml), and a 405 nm laser was used. Between samples, the instrument was flushed with 10% ethanol and distilled water. Reads of 1 min duration were performed in five replicates for each sample with the following capture settings: cell temperature: 25 °C, syringe load/flow rate: 30, camera: sCMOS. For capture settings, camera level was adjusted so that all particles were distinctly visible (Camera level 12–15). The ideal detection threshold was set including as many particles as possible and debris (blue cross count) with a maximum of five (detection threshold 5). Data output was acquired using NanoSight NTA software version 3.3 (Malvern Instruments). For each sample, the mean particle number in the Experiment Summary output was adjusted by the dilution factor.

**Protein, DNA, and RNA content.** As previously described[83–87], quantification of vesicle content, including protein, DNA, and RNA, was conducted using the Qubit® Protein Assay, Qubit® dsDNA high sensitivity assay, and RNA high sensitivity assay kits, respectively (Thermo Fisher Scientific). Protein, DNA, and RNA measurements were performed using a Qubit® 4 or Qubit® 3 Fluorometer. Instructions of the manufacturer were followed. There was no pre-treatment with DNase or RNase prior to the measurements.

**Lipid content.** The quantification of lipid content in AEVs was conducted using the FM4-64 lipophilic fluorescent dye and a linoleic acid standard, a method previously employed for bacterial extracellular vesicle (BEV) lipid quantification[88]. The modified procedure for quantifying vesicles released in culture was previously described in Juodeikis et al.[59] and includes the following steps: In duplicate, 20 µl of 30 µg/ml FM4-64 (Thermo Fisher Scientific) was combined with 180 µl of filtered culture supernatant or a linoleic acid standard in water (100; 75; 50; 20; 10; 5, 1; 0 µg/ml, prepared from a 1 mg/ml stock) in black 96-well plates. Following a 10 min incubation at 37 °C, endpoint fluorescence was analyzed using the FLUOStar Omega microplate reader with pre-set FM 4–64 settings (Excitation: 515-15; Dichroic: auto 616.2; Emission 720-20), employing an enhanced dynamic range. Linear standard curves from the linoleic acid samples were established for lipid quantification.

## Proteomics

Protein profiles of whole cell lysates (WCL) and AEVs were analyzed. 20 mg of cell biomass (3 replicates per species) were subjected to extensive ultrasonication with 400 μl of PBS. Cell debris was removed with centrifugation at $800 \times g$ at 4 °C, for 5 min. The supernatants were collected for proteomic analysis. The protein content of the whole cell lysate was determined by Pierce BCA protein assay according to the manufacturer's protocol (Thermo Fisher Scientific). Protein concentration of AEVs was measured by Qubit® Protein Assay (Thermo Fisher Scientific), as described above.

## Mass spectrometry analysis

For LC-MS/MS analysis, 2 (for AEVs) or 5 μg (for WCLs) of protein was reduced and alkylated for 10 min at 95 °C with final 10 mM TCEP (tris(2-carboxyethyl)phosphine) and 40 mM CAA (2-Chloroacetamide). The sample was processed according to the SP3 protocol[89] and digested overnight with trypsin (Promega, enzyme/protein 1:50). Peptides were desalted using SBD-RPS tips as previously described[90]. 400 ng per sample (re-dissolved in 2% acetonitrile/0.1% formic acid in water) was subjected to LC-MS/MS analysis. Protein digests were separated by nano-HPLC (Dionex Ultimate 3000, Thermo Fisher Scientific) equipped with a C18, 5 μm, 100 Å, 100 μm × 2 cm enrichment column and an Acclaim PepMap RSLC nanocolumn (C18, 2 μm, 100 Å, 500 × 0.075 mm) (all Thermo Fisher Scientific). Samples were concentrated on the enrichment column for 5 min at a flow rate of 15 μl/min with 0.1% formic acid as isocratic solvent. Separation was carried out on the nanocolumn at a flow rate of 300 nl/min at 60 °C using the following gradient, where solvent A was 0.1% formic acid in water and solvent B was acetonitrile containing 0.1% formic acid: 0–5 min: 2% B; 5–123 min: 2–35% B; 123–124 min: 35–95% B, 124–134 min: 95% B; 134–135 min: 2% B; 135–150 min: 2% B. The maXis II ETD mass spectrometer (Bruker Daltonics) was operated with the captive source in positive mode with the following settings: mass range: 200–2000 m/z, 2 Hz, capillary 1600 V, dry gas flow 3 L/min with 150 °C, nanoBooster 0.2 bar, precursor acquisition control top 20 (collision induced dissociation (CID)). Full mass spectrometry proteomic data were deposited to the ProteomeXchange Consortium[91] via the partner repository with the dataset identifier PXD053245[91].

The LC-MS/MS data were analyzed by MSFragger[92,93] by searching the public *Methanobrevibacter* protein databases (UP000232133; UP000003489; UP000004028; UP000018189; UP000001992), the archaeal protein catalog described in Chibani et al.[25] and a list of common contaminants[94]. Additional information on proteins found in all vesicles was retrieved via MaGe[40] and the implemented functions SignalP (version 4.1)[95], MHMM (version 2.0c)[96,97] and InterProScan[31,98], as well as from the InterPro Database[31] (Supplementary Data 7).

Carbamidomethylation of cysteine and oxidation on methionine were set as a fixed and as a variable modification, respectively. Detailed search criteria were used as follows: trypsin, max. missed cleavage sites: 2; search mode: MS/MS ion search with decoy database search included; precursor mass tolerance ±20 ppm; product mass tolerance ±15 ppm; acceptance parameters for identification: 1% protein FDR[99].

Data from EV and whole cell lysates were processed with Perseus software version 1.6.15.0. Data was filtered for decoy hits and contaminants. After log2 transformation and subtracting the median from the column proteins were filtered for containing at least 2 valid values in at least one group.

## Cell fractionation for proteomics analysis

100 mg of biomass from *Methanobrevibacter smithii* ALI and *M. intestini* (five replicates per species) were subjected to extensive ultrasonication with 3000 μl of PBS. Cell debris was removed via centrifugation at $800 \times g$ at 4 °C. The supernatants were collected (whole cell lysate - WCL). A cytoplasmic fraction (CF) was prepared from these whole cell lysates by centrifugation at $100.000 \times g$ for one

hour at 4 °C. The pellet was resolubilized with 400 μL PBS containing 1% SDS to collect the membrane fraction (MF). The protein content was determined by BCA assay according to the manufacturer's protocol (Thermo Fisher Scientific, USA).

For LC-MS/MS analysis 50 μg of precipitated protein were resolubilized, reduced and alkylated 20 min at 60 °C with final 10 mM TCEP (tris(2-carboxyethyl)phosphine) and 40 mM CAA (2-Chloroacetamide) in 100 mM TrisHCl pH 8,5 containing 25% Trifluroethanol (TFE). After dilution to 10% TFE the samples were digested overnight with trypsin (Promega, enzyme/protein 1:50). Peptides were desalted using SBD-RPS tips as previously described[90]. 400 ng per sample (re-dissolved in 2% acetonitrile/0.1% formic acid in water) was subjected to LC-MS/MS analysis. Protein digests were separated by nano-HPLC (neoVanquish, Thermo Fisher Scientific) equipped with a C18, 2 μm, 300 μm × 5 mm enrichment column and an DNVnanocolumn (C18, 2 μm, 100 Å, 75 μm × 500 mm) (all Thermo Fisher Scientific). Samples were concentrated on the enrichment column at a flow rate of 15 μl/min with 0.1% trifluoroaceticacid as isocratic solvent via flow control. Separation was carried out on the nanocolumn at a flow rate of 300 nl/min at 60 °C using the following gradient, where solvent A is 0.1% formic acid in water and solvent B is 80% acetonitrile containing 0.1% formic acid: 0–45 min: 3–23% B; 45–60 min: 23–40% B, 60–60.1 min: 40–99% B; 60.1–70 min: 99% B. The Orbitrap Exploris mass spectrometer (Thermo Fisher Scientific) was operated with the nanoFlex source and FAIMS in positive mode with the following settings: mass range: 350–1200 m/z, Spray voltage: 2000 V, carrier gas flow 3.8 L/min, ion transfertube 275 °C. MS data were acquired in data-dependent acquisition (DDA) mode, dynamic exclusion on, with following settings: MS1 resolution: 60 K, MS2 resolution: 15 K. FAIMS CV voltages: alternating – 45 V and – 65 V. controlled via cycle time: 1.5 sec per CV.

The LC-MS/MS data were analyzed by MSFragger[92] by searching the public *Methanobrevibacter* protein databases (UP000232133; UP000003489; UP000004028; UP000018189; UP000001992) including an in-house species-specific database and common contaminants. Carbamidomethylation on cysteine and oxidation on methionine were set as a fixed and as a variable modification, respectively. Detailed search criteria were used as follows: trypsin, max. missed cleavage sites: 2; search mode: MS/MS ion search with decoy database search included; precursor mass tolerance ±20 ppm; product mass tolerance ±15 ppm; acceptance parameters for identification: 1% protein FDR[99]. Additionally, a label free quantification (LFQ) was performed for the whole cell lysates using IonQuant[100] requiring a minimum of two ratio counts of quantified peptides.

Data was processed with Perseus software version 2.0.10.0. Data was filtered for decoy hits and contaminants. After log2 transformation proteins were filtered for containing min. 4 valid values in at least one group. The missing values were imputed with random numbers that are drawn from a Gaussian distribution. The values were optimized to simulate a typical abundance region that the missing values would have if they had been measured.

## Mass spectrometry derived AEV metabolomics

Biological triplicates of the vesicle preparations were used for the LC-MS analysis, and a technical duplicate of a non-cultured medium that had passed through the pipeline for vesicle isolation was used as a medium blank. All samples were stored at −70 °C until processing at the Vienna BioCenter Metabolomics Core Facility.

The samples were diluted with 50 μl ACN and subjected to analysis with liquid chromatography-mass spectrometry (LC-MS). 11 μl of each sample was pooled and used as a quality control (QC) sample. Samples were randomly injected on an iHILIC®-(P) Classic HPLC column (HILICON AB, 100 × 2.1 mm; 5 μm; 200 Å) with a flow rate of 100 μl/min delivered through an Ultimate 3000 HPLC system (Thermo Fisher Scientific). The stepwise gradient has a total run time of 35 min, starts at 90% A (ACN), and takes 21 min to 60% B (25 mM ammonium

bicarbonate) followed by 5 min hold at 80% B and a subsequent equilibration phase at 90%. The LC was coupled to a high-resolution tandem MS instrument (Q-Exactive Focus, Thermo Fisher Scientific). The ionization potential was set to +3.5/−3.0 kV, the sheet gas flow to 20, and an auxiliary gas flow of 5 was used. Samples were flanked by a blank and a QC sample for background labeling and data normalization, respectively.

The obtained data set was processed by "Compound Discoverer 3.3 SP2" (Thermo Fisher Scientific). Annotation of the compounds was done through searching against our internal mass list database generated with authentic standard solutions (highest confidence level). Additionally, the mzCloud database was searched for fragment matching and ChemSpider hits were obtained using BioCyc, Human Metabolome Database, E. coli Metabolome Database, and KEGG databases. Only metabolites identified with highest confirmation (match with internal database) were examined in more detail; additional ones are provided in Supplementary Data 10.

The log2 fold changes, as well as p-values, were calculated by the Compound Discoverer software (Tukey HSD test post hoc, after an analysis of variance (ANOVA) test).

Full mass spectrometry metabolomic data were deposited to MetaboLights[101] with the dataset identifier MTBLS12422[101].

## Co-incubation experiments with cell lines

**HT-29 and THP-1 cultivation and Co-incubation.** The HT-29 co-incubation experiments were performed at the Institute of Molecular Biosciences, University of Graz, Graz, Austria. HT-29 (intestinal epithelial cells) were grown in T-175 tissue culture flask, containing Dulbecco's Modified Eagle's medium/Nutrient F-12 (DMEM-F12) medium (Gibco) supplemented with 10% fetal bovine serum (FBS), penicillin-streptomycin (100 µg/ml streptomycin and 100 Units/ml penicillin) and L-glutamine (2 mM) at 37 °C in a 5% $CO_2$ incubator. To investigate the immunostimulatory potential of AEVs and BEVs, HT-29 cells were seeded in 24-well tissue culture plates at a concentration of 6 ×$10^5$ cells/well and cultivated for 24 h in DMEM-F12 medium supplemented with 10% fetal bovine serum (FBS), penicillin-streptomycin and L-glutamine. Then the intestinal epithelial cells were washed once with PBS and the medium was replaced with AEVs or BEVs ($10^8$ and $10^9$ particles/ml cell culture medium) resuspended in DMEM-F12 medium without FBS. After incubation for 20 h, the cell culture supernatant was harvested, centrifuged at 2500 rpm at 4 °C for 10 min to remove the cell debris, and stored at −20 °C for Luminex analysis. AEVs from each archaeal strain were incubated in triplicates, triplicates were also used for Luminex analysis.

The THP-1 co-incubation experiments were performed at the Core Facility Alternative Biomodels and Preclinical Imaging, Medical University of Graz. THP-1 cells were cultured in RPMI 1640 (Gibco, Life Technologies) supplemented with 10% fetal bovine serum, 2 mM L-glutamine and 1% penicillin (10,000 U/ml) / streptomycin (10,000 U/ml) (Gibco, Life Technologies) in a humidified incubator set to 5% $CO_2$ atmosphere at 37 °C. Prior co-incubations with bacterial and archaeal EVs, 2 × $10^5$ cells were differentiated to macrophages by 24 h incubation with 150 nM phorbol 12-myristate 13-acetate (PMA, Sigma, P8139) at 37 °C in a 5% $CO_2$ atmosphere. After one and four days, medium (RPMI + 10% FBS) was changed. On day five, cells were treated with two concentrations of EVs ($10^8$ and $10^9$ particles/ml cell culture medium) and incubated for 24 h. Supernatant was collected for cytokine analyses and stored at −20 °C for Luminex analysis. For Luminex analysis, samples were pooled and measured in duplicates.

Both, HT-29 and differentiated THP-1 cells were exposed to bacterial EVs (ETEC and B. fragilis) and archaeal EVs (M. smithii ALI, M. intestini, M. smithii GRAZ-2, and M. stadtmanae).

**Cytotoxicity tests of AEVs and BEVs.** 3-(4,5-Dimethyl-2-thiazolyl)-2,5-diphenyl-2H-tetrazolium bromide (MTT) cell viability assays were

routinely performed at the end of the HT-29 cell culture assays[102], where HT-29 cells were co-incubated with bacterial EVs (originated from ETEC and B. fragilis) and archaeal EVs (originated from M. smithii ALI, M. intestini, M. smithii GRAZ-2, and M. stadtmanae). Also, a no treatment control (NTC) was included in the assay and three doses of EVs were tested: $10^7$, $10^8$, and $10^9$ particles/ml (Supplementary Fig. S6, Supplementary Data 13).

Additionally, CellTiter-Glo® 2.0 Cell Viability Assay (Promega) was used according to the manufacturer's protocol to investigate the cytotoxicity of AEVs from M. smithii ALI, M. intestini, M. smithii GRAZ-2, and M. stadtmanae on THP1-Blue™ (cells derived from THP-1 monocytes to monitor NF-κB signal transduction) cells, during the preparation for confocal microscopy and THP-1 cell culture assays. Two doses of EVs ($10^8$ and $10^9$ particles/ml) were tested for the THP-1 cell culture assays, and one dose ($10^{11}$ particles/ml) for confocal microscopy (Supplementary Fig. S6, Supplementary Data 13). As a comparison, a no treatment control (NTC) was included in the assay.

**Confocal microscopy.** M. smithii ALI, M. intestini, M. smithii GRAZ-2, and M. stadtmanae-derived AEVs ($1 × 10^{11}$/ml) were labeled with 5% (v/v) 3,3′-Dioctadecyloxacarbocyanine perchlorate (DiO) Vybrant cell-labeling solution (V22886; Thermo Fisher Scientific) at 37 °C for 30 minutes. Unbound dye was removed by washing with 3× PBS using centrifugal filters (100 kDa MWCO, Sartorius). The DiO - AEV suspension was isolated by SEC, checked for sterility then size and concentration determined using NTA, all as above. Labeled DiO - AEVs ($1 × 10^{11}$/well [10 µl]) were added to THP1-Blue™ cell monolayers cultured on collagen solution (Merck) coated 12-well chamber slides (IBIDI) overnight (16 hrs). THP1-Blue™ monocytes were previously induced to differentiate into macrophages by 24 h incubation with 150 nM phorbol 12-myristate 13-acetate (PMA, Sigma, P8139) at 37 °C in a 5% $CO_2$ atmosphere, followed by 24 h incubation in RPMI medium prior to addition of DiO - AEVs. Samples were fixed using Pierce 4% paraformaldehyde (PFA; Thermo Fisher Scientific), permeabilized with 0.25% Triton X1000 (Sigma), and blocked with 10% goat serum in PBS. For nuclear visualization, cells were incubated with Hoechst 33342 (Thermo Fisher Scientific) and Alexa 647-Phalloidin (Thermo Fisher Scientific) was added to visualize the cytoskeleton. In a second approach, AEVs were labeled with specific rabbit-derived polyclonal anti-archaea antibodies (1:1.000 dilution, anti-M. stadtmanae, Davids Biotechnologie GmbH) and subsequently stained with Alexa Fluor 647 (AF647) as a secondary antibody, in addition to DiO labeling. In this approach, THP1-Blue™ monocytes were incubated only with Hoechst 33342 (Thermo Fisher Scientific) for nuclear staining. Images were taken using a Zeiss LSM880 confocal microscope equipped with a 63 x /1.40 oil objective. Fluorescence was recorded at 405 (blue, nucleus), 488 (green, AEVs), and 594 nm (red, intracellular membranes or AEVs). The red channel was adjusted using the ZEISS ZEN 3.9 (ZEN lite) software by the best-fit function.

**Generation of archaeal antibodies.** Polyclonal antibodies against M. smithii and M. stadtmanae were generated at Davids Biotechnologie GmbH. Cell biomass of M. smithii or M. stadtmanae was used for the immunization of rabbits. A 63-day protocol was followed including 5 immunizations of the animal and an ELISA titer on day 35. For the experiments in this paper, the affinity purified antisera were used.

**Luminex analysis.** Supernatants from HT-29 and differentiated THP-1 cells exposed to bacterial and archaeal EVs were analyzed using a human cytokine 23-plex kit (R&D Systems) to assess the levels of 23 different cytokines as previously reported[48]: These included CCL2/MCP-1, CCL7/MCP-3, CCL20/MIP-3-alpha, CXCL1/GRO alpha, CXCL2/GRO beta, CXCL5/ENA-78, CXCL10/IP-10, granulocyte-macrophage

colony stimulating factor (GM-CSF), IL-17E/IL-25, tumor necrosis factor alpha (TNF-alpha), CCL5/RANTES, CXCL9/MIG, CXCL11/ITAC-1, CCL11/eotaxin, gamma interferon (IFN-gamma), CXLC8/IL-8, IL-18, IL-6, CX3CL1/fractaline, TSLP, CCL22/MDC, CCL28/MEC, and TGF-beta1. 50 μl of frozen cell culture supernatant samples were processed in 96-well plates according to the manufacturer's instructions. Standard curves for each analyte were generated using the reference analyte concentration provided by the manufacturer.

The measurement was performed on a calibrated Bio-Plex 200 system (Bio-Rad) and analyzed with Bio-Plex Manager software (version 6.1, Bio-Rad, Supplementary Data 11). Cytokine concentrations were determined from standard curves using five-parameter logistic (5PL) curve fitting. Supernatants from HT-29 cells mock treated with saline and THP-1 cell supernatants mock treated with DMSO served as negative controls. From the measured values, the mean was calculated and the negative controls were subtracted as blanks from samples. Values are displayed as log10 transformed (Fig. 6), if subtraction of negative controls lead to negative values, these were multiplied by (−1) to obtain positive values for log10 transformation, and were then multiplied by (−1) again to display negative values in the heatmap (Fig. 6, Supplementary Data 12).

### Statistics and data visualization

Vesicle properties (concentration, size, nucleic acids, and protein content) and metabolites were plotted as boxplots in R (R-Core-Team, 2024) using the ggplot2 Package (v3.5.1)[103] and finalized in InkScape[104]. Statistics were calculated in IBM SPSS Statistics (v. 29.0.0.0)[105]. The overview of proteins identified in archaeal vesicles and whole cell lysates, proteins annotated as adhesins, as well as the Luminex immunoassay were displayed in heatmaps using ggplot2 (v3.5.1)[103], with data transformation performed using the reshape2 package (v1.4.4)[106], and finalized in InkScape[104]. Bar chart of mean intensities/relative abundances of protein categories was plotted with ggplot2 (v3.5.1)[103], and dplyr (v1.1.4)[107] was used for the calculation of mean and standard deviation, and finalized in InkScape[104]. Metabolites were visualized as jitter plots using ggplot2 (v3.5.1)[103], and dplyr (v1.1.4)[107]. Workflow of vesicle isolation was created with the online tool draw.io (v26.0.10.)[108]. Creation of Venn diagrams was performed by using the online tool interactiVenn[109]. PCA was created with Perseus software (v1.6.15.0)[110]. Bar charts of selected protein groups and cell viability tests of HT-29 and THP-1 cells were created using ggplot2 (v3.5.1)[103], dplyr (v1.1.4)[107], and finalized in InkScape[104]. Tidyverse (v2.0.0)[111] and rstatix (v0.7.2)[112]. Statistically significant differences between two values were considered when *$p < 0.05$, **$p < 0.01$, and ***$p < 0.001$. ChatGPT (OpenAI) was used to improve the clarity and grammar of the manuscript.

### Reporting summary

Further information on research design is available in the Nature Portfolio Reporting Summary linked to this article.

## Data availability

Full proteomic data obtained from mass spectrometry are available via ProteomeXchange Consortium partner repository under dataset identifier PXD053245. Metabolomic data used in this study are available via MetaboLights under accession number MTBLS12422. Other data generated in the study are provided in the Supplementary Information, Source Data files and/or our Github repository (https://github.com/vikwein/Archaeal_extracellular_vesicles). Source data are provided with this paper.

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

## Acknowledgements

We thank Stefanie Duller for providing electron micrographs, Eliska Sedlackova for additional proof reading, and the John Innes Centre Bioimaging facility and staff for their contribution to this publication. The support for V. Weinberger through the local dissertation program MolMed is acknowledged. This research was funded in whole or in part by the Austrian Science Fund (FWF) [10.55776/F83, 10.55776/P32697, and 10.55776/COE7]. For open access purposes, the author has applied a CC BY public copyright license to any author accepted manuscript version arising from this submission. The authors acknowledge the support of the ZMF Galaxy Team: Core Facility Computational Bioanalytics, Medical University of Graz, funded by the Austrian Federal Ministry of Education, Science and Research, Hochschulraum-Strukturmittel 2016 grant as part of BioTechMed Graz. The Vienna Bio-Center Core Facilities (VBCF) Metabolomics Facility acknowledges funding from the Austrian Federal Ministry of Education, Science & Research; and the City of Vienna.

## Author contributions

The study was designed by C.M.E. and V.W. V.W., P.M., and T.Z. isolated the vesicles, together with help from R.S., E.J., and S.R.C. Vesicle biophysical characterization was done by V.W. V.W. and B.D. performed proteomics and analyzed the data with the supervision of H.K. and C.M.E. Metabolomics was performed by T.Koe and G.G., and data were analyzed by V.W. and C.M.E. Electron microscopy was performed by D.P., K.H., D.K., and K.G. H.T. and S.S. performed experiments with HT-29 cells. C.Ka and B.R. performed the co-incubation experiments with differentiated THP-1 cells. The immunoassay was planned by H.S. and performed by J.O. Confocal microscopy of macrophages was performed by V.W., with the help of R.S. and E.J. The lipid assay was performed by R.J. V.W. and C.M.E. wrote the manuscript, and T.S., C.Ku., R.M., T.Kue., and T.W. contributed to the writing of the manuscript and figure preparation. The manuscript was read and approved by all authors.

## Competing interests

The authors declare no competing of interests.

## Additional information

[1]Diagnostic and Research Institute of Hygiene, Microbiology and Environmental Medicine, Medical University of Graz, Graz, Austria. [2]Core Facility Mass Spectrometry, Medical University of Graz, Graz, Austria. [3]Institute of Molecular Biosciences, University of Graz, Graz, Austria. [4]Food, Microbiome and Health Institute Research Programme, Quadram Institute Bioscience, Norwich, United Kingdom. [5]Vienna BioCenter Core Facilities GmbH, Metabolomics, Vienna, Austria. [6]Core Facility Alternative Biomodels & Preclinical Imaging, Medical University of Graz, Graz, Austria. [7]Core Facility Flow Cytometry, Medical University of Graz, Graz, Austria. [8]Core Facility Ultrastructure Analysis, Medical University of Graz, Graz, Austria. [9]Advanced Microscopy Facility, Quadram Institute Bioscience, Norwich, United Kingdom. [10]Center for Pathobiochemistry and Genetics, Medical University of Vienna, Vienna, Austria. [11]Norwich Medical School, University East Anglia, Norwich, United Kingdom. [12]Field of Excellence Biohealth – University of Graz, Graz, Austria. [13]BioTechMed, Graz, Austria. ✉e-mail: christine.moissl-eichinger@medunigraz.at

