## [Transparent Peer Review file · Nature Communications]

Proteomic and metabolomic profiling of extracellular vesicles produced by human gut archaea

Corresponding Author: Professor Christine Moissl-Eichinger

Version 0:

Reviewer comments:

Reviewer #1

(Remarks to the Author)

In this study, the authors characterized the human archaeal EVs, using proteomic and metabolomic analyses. The authors found that EVs from *M. smithii* ALI and *M. intestini* are rich in adhesins compared to those from whole cells. The AEVs were also significantly loaded with amino acids and could be the neurotransmitters. Furthermore, the AEVs were taken up by a macrophage and induced IL-8 secretion by HI-29 cells, suggesting that the AEVs may play a role in host-archaea interactions. The experiments are well designed, and the results are well organized. On the other hand, EVs from human gut microbes have been well characterized in several papers. This manuscript focuses on archaeal EVs, but there are no surprising findings in this study and it lacks impact. Although proteomic and metabolomic analyses were performed, a detailed analysis of the factors contained in EVs was not performed. My specific comments are as follows.

Major comments

1. It is unclear what is a strong message in this study. Although this study is a first report to characterize EVs derived from methanogenic archaea, there is no clear indication of what interesting phenomena distinguish them from other bacterial or archaeal EVs. If the characteristics of EVs are similar to those of other microbes, it is preferable to publish in a microbiology journal rather than a general journal such as Nat Commun.
2. The authors concluded that adhesins/adhesin-like proteins are highly enriched in EVs, but these seem to be localized on the bacterial surface or extracellularly. The enrichment of these proteins in EVs compared to the whole cell lysates is observed as a matter of course. Are these proteins more concentrated in the EVs as compared to the cell membrane fraction? Are they not present in extracellular supernatants (removed EVs)?
3. Similar concerns as above should be noted for the metabolomics results. Although glutamic acid and aspartic acid are enriched in EVs compared to the control medium, it remains unknown whether these amino acids are concentrated in EVs or whether they are also abundant in the supernatant. The authors need to investigate whether they are specifically present in EVs.
4. To investigate the pro-inflammatory potential of archaeal EVs, the authors examined only the IL-8 cytokine response. For a comprehensive understanding of the immune response in epithelial cells, other cytokine production should also be examined. Although a certain concentration of EVs was examined in the current manuscript, it is necessary to investigate whether these are concentration-dependent responses.
5. The induction levels of IL-8 secretion are significantly different among species-derived EVs, but is this due to the presence of adhesion? The relationship between the presence of adhesion and the production of IL-8 is not clear at all.

Other comments

6. Figure 1: Since only one vesicle of *M. intestini* is shown in the TEM images, several vesicles should be shown.
7. Please italicize the strain name appropriately in the article and in figures and legends. Also, the font size is too small in all figures, please make the figures larger. The number of biological replicates and measurements should be included in the figure legends.

8. Figure 2: It appears that the number of dots is different between strains. Does the number of dots on the graphs indicate the number of measurements or independent biological samples? In the legends (line 165), the criteria (threshold) for the values should be clearly stated.
9. Line 202: Please include "Principal component analysis" in the text before using the abbreviation.
10. Figure 4 (B): What does "intensities" mean? Also, why is the order of the groups shown in the visualizations different between *M. smithii* and *M. intestini*? If there is no reason, please align them.
11. Line 248-251, Please include figure citations as appropriate in the text. For example, "Within a group of transport-associated proteins, we found substantial enrichment of a protein (representative: GUT_GENOME043902_01504) with an OPT (oligopeptide transporter) superfamily domain (Fig. 4B), which...".
12. Fig. 4B: Show the intensities of proteins related to ①, ②, and ③ compared with those of whole cells, or heatmap like Fig. 4D for ①, ②, and ③. It is important to focus on proteins that are more abundant in EVs than in cells.
13. Figure 5: Box and whisker plots are not appropriate because the authors only plotted three biological replicates.
14. Line 291: In the case of "significantly changed", please indicate the calculation method and P-values.
15. The legend of Figure 6: please use EVs or AEVs consistently.
16. Figure 7: The bar graph must also include independent data (as shown by the dots). Why did the authors use "median and interquartile range", instead of "means \pm standard errors"?
17. Line 331-333: The induction of IL-8 secretion by *M. intestini* EVs is not at the same level as by ETEC EVs.
18. Line 368-369: The authors describe that "All vesicles were found to be taken up by human monocytes (Fig. 6) and to stimulate IL-8 secretion (Fig. 7, $P > 0.05$)", but only *M. intestini* EVs significantly induced IL-8 secretion. And what does this P value " $P > 0.05$ " mean?
19. Line 531-532: Yeast vesicles seem to pass through filters.

Reviewer #2

(Remarks to the Author)

Weinberger et al present a study on the capability of the archaeal genera *Methanobrevibacter* and *Methanosphaera* to produce extracellular vesicles (AEVs). The authors employed electron microscopy, proteomics and metabolomic analyses to characterize the properties of these AEVs. Furthermore, the authors explored the pro-inflammatory potential of these AEVs against human epithelial cell line.

While the study addresses an important area of archaeal biology, several aspects require further refinement. The evidence supporting the involvement of adhesin and adhesin-like proteins in archaea-surface and archaea-bacteria interactions, as well as the proposed connection between the identified glutamic acid and choline glycerophosphate with gut-brain signaling, appears to be insufficiently substantiated. I recommend that the authors consider tempering the confidence with which these conclusions are presented. Additionally, the overall clarity of the manuscript would benefit from significant improvements to ensure a coherent and strong logical flow throughout the text, and there are a few typo errors that needed to be checked and addressed. The resolution and readability of the figures should also be enhanced, as the current quality, particularly the small and blurred text within the figures, diminishes their effectiveness in conveying key points.

Specific comments are outlined below:

Line 29 and Line 32. Please correct "M. simithii, Cand. M.intestini" & "M. simithii and Cand. M.intestini"

Line 32. Please clarify which the "unique traits" are being referred to and how they are relevant to the study's findings.

Line 33. It is not clear whether "n=229" represents the total proteins identified in the EVs of both *M. simithii* and *M. intestine*, or it refers to the proteins found in the EVs of *M. intestini*. There is no explanation for why only proteomic analyses were performed for the EVs of *Methanobrevibacter* and not for *Methanosphaera*.

Line 37. Please clarify what IL-8 is and specify the purpose of using HT-29 cells, and explicitly state what the results involving IL-8 and HT-29 suggest.

Line 64. The authors should provide more detailed information about explosive membrane vesicles, including where they are produced and how they differ from outer membrane vesicles. A clearer explanation of these distinctions would enhance the reader's understanding of their relevance to the study.

Line 89. Please specify what is meant by the "defensive function" of these vesicles. Defensive against what?

Line 93. I think "Trough" is meant to be "Through".

Line 114. Supps Figure 1 should be cited for "AEV biomass production and an isolation protocol"

Line 121-122. Did the authors observe any significant difference in the sizes of vesicle-like structures within the cells compared to the isolated purified vesicles? A proper quantitative analysis should be performed to substantiate any conclusions regarding size differences. From the figures, it appears that the vesicle-like structures within the cells are generally smaller than the isolated vesicles. Could the authors provide comments or additional analysis on this?

Line 123. The "culture media controls" is not clearly defined. Please clarify whether specific media was used to introduce vesicle-like structures and provide data demonstrating that no vesicles were observed in the culture media controls.

Line 129. From electron microscopy images, it would be more accurate to refer to these structures as "vesicle-like structures" unless further evidence explicitly confirms their identify as AEVs.

Line 137. Table 1 could be moved to the Supplementary Material. Additionally, please specify the number of biological replicates performed in the figure and table legends. The proper statistical analysis should be performed.

Line 142-143. References are required for "enterotoxigenic Escherichia coli (ETEC, ~120 nm)", as well as "B. fragilis (~194 nm)", respectively.

Line 145. Please specify the source of the data of ETEC (6.38E+11 particles/ml), and B. fragilis (8E+11 particles/ml).

Line 146. The statement "The concentrations were reasonably consistent for all M. smithii strains". All" should be revised to "both" if only two strains were analyzed. Please provide details on how this conclusion was reached. Proper statistical analysis should be performed to determine if there are significant differences or no differences between the strains.

Line 153 "the lowest amount of lipids (~4.9 µg/1010 particles on average) for which strain?

Line 163. The figure legend should be expanded to include more details. It should indicate the number of biological replicates, explain what the line inside the box represents, and describe the corresponding histogram and clarify the range of values depicted by the box.

Line 167. The table shows that the min and max values for lipid content are the same for M. intestini, M. smithii GRAZ-2, and M. stadtmanae. Please provide an explanation for this observation in the table legend.

Line 171. It is not clear why the protein profiles of EVs were analyzed only for M. smithii AL1 and M. intestini. Please clarify why the other two strains were not included in the analysis.

Line 181. Supplementary Figure S2 is not readable. The figure legend should specify the origin of the 394 proteins mentioned. There is also no description of these 394 proteins in the main text.

Line 182. Is there a specific reason for comparing BEVs from B. thetaiotaomicron in this context, rather than including BEVs from other species? Providing a rationale for this choice will help clarify the relevance of the comparison.

Line 190. The Figure 3 could be moved to Supplementary Material. The figure is difficult to read due to the use of many abbreviations. Consider including species names in the diagrams, such as M. smithii AL1_EV. There appears to be sufficient space in the figure to for these labels to improve clarity. Having many abbreviations in the main text affects readability.

Line 208. The same comment as above. There is no need to have so many abbreviations.

Line 215. The description of "3+3 biological replicates" is not clear.

Line 232-236. Strictly speaking, are these really domains (Invasin/intimin cell-adhesion fragments; Pectin_lyase_fold/virulence)? Does InterPro call these domains?

Line 237. "Bacterial proteins containing Ig-like domains exhibit a broad spectrum of functions," This sentence and the paragraph that follows seems to indicate that Ig-like "domains" and pectin lyase folds are found in such diverse proteins that their functon(s) in adhesin-like proteins cannot be inferred with any confidence.

Line 240-242. Reference is required

Line 243-245. Reference is required.

L249-251. "OPT (oligopeptide transporter) superfamily domain, which in prokaryotes may contribute to iron-siderophore uptake, indicating a potential role in iron binding"
The meaning here isn't entirely clear. Is this OPT predicted to import oligopeptides or complexed iron or both?

L260-262. Is this free glutamic and aspartic acid, or part of proteins/polypeptides contained in the EV? This is not clear.

Line 262. It is not clear whether “a substantial Log2 Fold change (FC)” refers to changes in abundance or another measurement. The “background samples” is not clearly defined. Please also explain how the conclusion that “these amino acids are important cargos for both species” was reached.

Line 301. Please specify whether DiO was used to label the membrane of AEVs.

Line 309 and 313. Please specify which microscope was used to capture these images in the figure legend. Panel E shows an orthogonal slice image rather than a representative Z-stack. Please label the X-Y, Y-Z, and X-Z planes in the figure. To better display the results, please place Z-slice images at different Z-depths on the right side.

Line 343. How did the authors ensure that the particle amounts of AEVs were equal? Additionally, could the size of the AEVs affect the results? Please address these concerns to clarify how the study controls for these variables?

Line 371-373. "Methanobrevibacter species rely on syntrophic bacterial partners that provide small organic compounds like H₂ (or formate) and CO₂ for methanogenesis"

It might not be correct to refer to H₂ as organic, even if (as here) it has a biotic origin.

Line 383. "From the bacterial kingdom"--domain, not kingdom.

Line L385 "Considering that adhesins are highly enriched in AEVs"

The proteins mentioned earlier are not demonstrated to be adhesins, but adhesin-like proteins. A function in adhesion has yet to be determined, and the presence of Invasin/intimin cell-adhesion fragments and Pectin lyase fold does not necessarily point to a function in adhesion.

L388-389 "In Methanobrevibacter ruminantium, a prevalent Methanobrevibacter species in ruminants, 5% of the genome encodes adhesins"

The functions of almost all of these proteins as adhesins has yet to be demonstrated. Mru_1499 is exceptional in that it has been investigated experimentally.

Reviewer #3

(Remarks to the Author)

The manuscript focuses on the characterization of extracellular vesicles (EVs) isolated from archaeal strains that inhabit the human gut. In addition to the proteomic and metabolic profiling of the isolated vesicles, authors conduct functional studies in macrophages and intestinal epithelial cell lines.

The study of archaeal extracellular vesicles (AEVs) is a relevant research area in the context of the gut microbiota. In the last years, research on the gut microbiome has been almost exclusively focused on bacteria. In this context, studies on other microbiota components such as viruses, fungi and archaea are needed to understand their contribution to intestinal homeostasis and human health.

The topic is interesting and although the general approach is fine, there are some questions and comments that should be addressed:

1. The first paragraph of the results section (lines 110-116) is just a summary of the following result subsections. It could be removed, and the information introduced into each specific subsection.

2. Figure 1 shows EVs inside the cells. How can authors prove that these are intracellular vesicles that originate the EVs? Which is the mechanism of EV formation in these strains? Did the proteomic analysis identify ESCRT proteins in the EVs? This information may shed light on the potential mechanism of vesicle biogenesis in intestinal archaeal strains.

3. Figure 1 and Figure 3: Lettering and legends are illegible even in the original figures (big size).

4. Table 1. There is a great variability in protein and DNA concentration data from different batches of EVs isolated from a single strain (as an example, protein concentration ranges from 0.09 ug/10¹⁰ particles to 180 ug/10¹⁰ particles). In the discussion (lines 359-366), the authors point to growth conditions (culture phase and medium composition) as main factors influencing the vesicle size, amount and cargo. In this sense, EV samples collected from cultures carried out with the same strain, medium, and conditions should be more homogeneous. This is relevant since data presented in table 1 are normalized to 10¹⁰ particles. Normalization by particle number should yield more similar values.

5. Lines 181-183: Authors compare the number of proteins identified in archaeal EVs with the number of proteins identified in BEVs from *B. thetaiotaomicron* and conclude that the number of proteins in archaeal EVs is lower. Which is the reasoning for this comparison? The number of vesicular proteins identified depends on the methodology used, data curation, the producer strain, and the growth conditions. In reference 57, the EV analyzed were collected from in vitro BHI cultures and from in vivo samples. Moreover, I am not sure whether the 2,047 proteins indicated in the manuscript (lines 181-182) are

indeed proteins identified in *B. thetaiotaomicron* EVs or in the parental cells after data curation (see reference 57, page 11, section "Proteomics Data Curation").

6. I am not expert in metabolomics, but the analysis has identified very few metabolites. Please discuss.

7. The metabolomic profiles indicate that EVs from *M. smithii* Ali and *M. intestine* are enriched in glutamate and aspartate. Did authors check whether the high amount of these amino acids in EVs correlate with proteomic data showing the presence of metabolic enzymes that catalyze their production (such as asparaginase) as vesicle cargo?

8. Why authors study internalization of AEVs in THP1-derived macrophages but analyze their pro-inflammatory effect (IL-8 secretion) in HT29 epithelial cells?

Macrophages are phagocytic cells, so they can take up vesicles by means of this activity, whereas intestinal epithelial cells, which are non-phagocytic cells, usually internalize EVs through endocytosis. Immunofluorescence microscopy analysis of labelled vesicles in HT29 cells would provide knowledge on the internalization pathway of AEVs in epithelial cells.

Concerning the inflammatory response, why not analyze cytokine secretion by macrophages in response to AEVs? In fact, macrophages are immune cells.

Internalization studies in HT29 cells and cytokine analysis in THP1 derived macrophages would complete the functional part of the manuscript.

9. Lines 300, 678: what does the letter b in THP1-b mean?

10. Please provide details on the PMA protocol used to differentiate THP1 monocytes into macrophages (incubation time with PMA and days of rest) in the confocal microscopy section (line 679).

11. Lines 531-532: Which filter size have authors used to eliminate the vesicles and other residues from the yeast extract used for medium preparation? Vesicles smaller than 200 nm are not eliminated using a 0.22 µm filter.

12. Line 358: "...although the particle count was substantially lower for archaea (Fig 2., Table 1)". Table 1 and Figure 2 do not present data for bacterial EVs.

13. Minor comments:

- Line 11 and abstract: the strain names should be in italics.

- Line 447: In the following, we will use the abbreviation *M. intestini* instead of *Candidatus M. intestine*. Since the Methods section is at the end of the manuscript (after discussion) this information should be provided the first time the strain appears (in the results section) to help the reader.

- Lines 455-456: the sentence lack the verb.

- Line 369: ...and stimulate IL-8 secretion (Fig 7...). Please, add in intestinal epithelial cells after secretion.

- Line 378: communication vehicles or molecules? The sentence refers to adhesins...

Reviewer #4

(Remarks to the Author)

The manuscript entitled "Proteomic and metabolomic profiling of archaeal extracellular vesicles from the human gut" is a strong foundational study analyzing the vesicle content and basic immune stimulatory ability of EVs derived from two biologically important archaeal species. The authors present a thorough and detailed analysis of EVs produced by archaeal species commonly present in the gastrointestinal tract. The importance of archaea in the overall function of the microbiome and health of the host underpins the significance of the presented work as the mechanisms through which these microbes communicate with each other and the host are poorly understood. The robust nature of the vesicle data presented and the stringent thresholds for vesicle analysis are among the strengths of this manuscript. While interesting, the biological significance of the metabolite data is unclear given that these bacteria are grown in nutritive media under laboratory conditions. However, the robust nature of the metabolome analysis does provide a strong foundation of information and a starting point for metabolomic analysis of host-derived vesicles. In contrast to the vesicle analysis data, the biological impact and cell response studies were not nearly of the same caliber. The authors give brief methods for cell viability assays (which admittedly are standard) but neglect to mention which AEV or BEVs are specifically tested with the cells and also do not mention what controls were used. Considering cell viability impacts immune responses, this data should be shown for all EVs used in IL-8 experiments. Dye only controls were also missing for EV entry experiments. In addition, the evaluation of only one cytokine as an indicator of immune stimulation is disappointing, particularly in light of the very thorough nature of authors vesicle analysis. Overall, the author's characterization of the vesicles is superb but there are concerns regarding the analysis of vesicle interaction with host cells. Major and minor concerns are listed below:

Line 93. Should the last word be "through"?

Line 145. Include representative NTA graphs as a supplemental figure.

Line 159. The authors do not mention in the methods if EVs were DNase or RNase treated prior to measuring DNA and RNA, respectively. If this was done, the methods used should be included in the methods sections. Extraneous, unencapsulated nucleic acids are commonly detected in EV preps and the lack of DNase/RNase treatment can inflate nucleic acid measurements. If these treatments were not done, experiments should be repeated or that caveat included in the results description and discussion.

Figures 2 and 4. The font size for axis legends and labels are impossible to read and need to be increased in size.

Line 174. The statement to "refer to materials and methods section" is unnecessary since that is always where such details are located. The statement should be removed.

Line 229. A reference to the publication showing that "rumen methanogens account for up to 5% of all genes" should be included.

Line 261. Since the bacteria are grown in vitro in nutritive media, it is unclear what the biological relevance of detected, packaged metabolites will be. Since substrates will be different in the gut, it stands to reason that packaged metabolites might also be different. Authors should address this in the discussion.

Line 302. Were dye controls used in this experiment? Dye aggregates regularly form with Dio and dye only controls that have been processed in parallel with (or at least in the same manner as) the OMVs and then applied to cells in parallel should be included to ensure that the signal observed is vesicles and not dye aggregates which are also endocytosed by macrophages. If this was not done, dye only controls should be tested to confirm that it is actually AEVs being taken up by the cells and not dye aggregates, particularly for larger fluorescent spots as dyed vesicles tend to produce punctate signals.

Line 398. It is unclear why this sentence is set aside from the paragraph above. It's grammatically incorrect to have a standalone sentence, so it should be combined with either the preceding or following paragraph.

Line 404. Which responses by HT-29? Assumed IL-8, but that should be specified since figures 6 and 7 are previously mentioned in the paragraph.

Line 408. The limitations of drawing conclusions from metabolomic data generated from EVs produced in culture media should be acknowledged in this paragraph.

Line 460. Could the authors elaborate on "prepared earlier"? How much earlier? There is a separate description of the methods used for BEV isolation, so the inclusion of this sentence in this methods section seems out of place and its purpose in this location unclear.

Line 517. If they were not processed immediately, how long were they stored at 4C? Considering long term storage results in degradation of EVs, the max amount of storage time is an important experimental consideration and should be included.

Line 530. Were protease inhibitors or any other diluents other than PBS used for isolation or final resuspension of AEVs?

Line 666. What AEVs were used in cell viability studies? What controls were used? Cell viability data should be included in supplemental section, particularly considering that cell viability can impact immune responses and differences in IL-8 stimulation are seen between AEVs and BEVs.

Line 701. How many EVs were actually applied to the HT-29 cells? The concentration is given but not the volume, which is needed for anyone wishing to reproduce these experiments. It has been shown that EV number matters when evaluating immune responses by cells, so this information is important for evaluation of the data.

Figure 7. The authors state n=6 but it is unclear based on the information included in the methods if that is 6 experimental replicates, 3 experimental replicates with 2 technical replicates, or some other technical/biological replicate combination. This should be clearly stated so the robustness of the result and analysis can be determined.

The detailed description of experimental methods was excellent, particularly those associated with vesicle analysis. A common problem in the field is lack of this information which hampers the ability to replicate methods between laboratories.

Reviewer #5

(Remarks to the Author)

Version 1:

Reviewer comments:

Reviewer #1

(Remarks to the Author)

Although the authors answered questions carefully, the current manuscript is still unlikely to be accepted for publication in Nature Communications.

Although I addressed that “Although proteomic and metabolomic analyses were performed, a detailed analysis of the factors contained in EVs was not performed.”, this issue has not yet been resolved in the revised manuscript. In particular, the title only states that the manuscript provides proteomic and metabolomic profiling, without clearly conveying what is novel and significant. This may be due to the lack of conclusive evidence identifying the key factors in the current research results. The authors rely solely on OMICS data and do not present definitive findings to support strong conclusions. If Adhesin or specific amino acids in EVs definitely play a critical role in gut cell linkage, further experiments will be needed to confirm this. There is a complete lack of data in the current manuscript to indicate what the impact would be in the absence (or increase or decrease) of adhesin or the specific amino acids. The title should specify which factors are important, rather than simply stating "Proteomic and Metabolomic Profiling."

With regard to each comment,

1. The newly added text aptly describes the importance of this study.
 2. The authors explained that adhesion is highly contained in EVs.
 3. The shown data is important and it is better to add into the supplemental results.
 4. The authors responded appropriately to the comments and the data obtained is important.
 5. I understand that IL-8 production is not determined by the presence of adhesin.
- For other minor comments, authors correctly answered and revised the manuscripts.

Reviewer #3

(Remarks to the Author)

The authors' efforts to address the reviewers' requirements were remarkable. The inclusion of new data and experiments has significantly strengthened the manuscript. Notably, the proteomic analysis of archaeal cell membrane fractions, which confirms the enrichment of adhesin-like proteins in the corresponding AEVs (as suggested by Reviewer 1), and the multiplex analysis of cytokines secreted by THP-1 macrophages and HT-29 cells (as recommended by multiple reviewers) have substantially enhanced the study.

The authors have addressed all my comments and suggestions from the initial manuscript review process, except for the internalization of AEVs in HT29 cells (item # 8). I disagree with the explanation provided in their response letter: *“Differentially to macrophages, epithelial cells are not expected to take up vesicles (at least not in a substantial amount, as a whole), so we did not perform experiments on the internalization of EAVs in epithelial cells”*.

Numerous studies with bacterial EVs (BEVs), either from Gram-negative and Gram-positive bacteria, have shown that epithelial cells efficiently internalize BEVs through endocytosis. Various endocytic pathways contribute to BEV uptake, depending on the vesicle size and cargo, including clathrin-mediated endocytosis, lipid raft- and caveolin-dependent endocytosis, micropinocytosis and even direct fusion membrane. The use of specific inhibitors targeting these pathways have proven invaluable in elucidating the mechanisms of BEV internalization. For further details, see for instance the review with doi: 10.1111/cmi.12655 and the article with doi: 10.1038/s41598-020-78920-z).

If uptake experiments in HT29 cells are not conducted, the authors should state in the manuscript that further studies are needed to characterize the internalization pathway in intestinal epithelial cells. The different profile of adhesins in AEVs, depending on the producer strain, may influence or determine the entry route.

Minor points:

Line 139: “... and attached to cells (Fig.1: a, e, K, l)...” The “K” should be “l”.

Line 204 (Figure 2 legend): “...A one-way ANOVA ($p=0.007$) followed by Tukey's HSD post hoc test revealed”. I believe that ($p=0.007$) should be removed here.

Reviewers Comments and Author Responses

We appreciate the time of all four reviewers and are thankful for their helpful comments and valid corrections.

Please find our responses below.

Reviewer #1:

In this study, the authors characterized the human archaeal EVs, using proteomic and metabolomic analyses. The authors found that EVs from *M. smithii* ALI and *M. intestini* are rich in adhesins compared to those from whole cells. The AEVs were also significantly loaded with amino acids and could be the neurotransmitters. Furthermore, the AEVs were taken up by a macrophage and induced IL-8 secretion by HI-29 cells, suggesting that the AEVs may play a role in host-archaea interactions. The experiments are well designed, and the results are well organized. On the other hand, EVs from human gut microbes have been well characterized in several papers. This manuscript focuses on archaeal EVs, but there are no surprising findings in this study and it lacks impact. Although proteomic and metabolomic analyses were performed, a detailed analysis of the factors contained in EVs was not performed. My specific comments are as follows.

Major comments

1. It is unclear what is a strong message in this study. Although this study is a first report to characterize EVs derived from methanogenic archaea, there is no clear indication of what interesting phenomena distinguish them from other bacterial or archaeal EVs. If the characteristics of EVs are similar to those of other microbes, it is preferable to publish in a microbiology journal rather than a general journal such as Nat Commun.

By focussing on bacterial vesicles in the introduction, we obviously did not make it clear enough why the discovery of AEVs from methanogens is an interesting result that should be reported to a wider audience.

We added the following paragraph: "The human archaeome stays an underexplored component of the microbiome, and this knowledge gap substantially limits our understanding of how archaea contribute to human health and disease. Unlike the extensive research on bacterial EVs, archaeal extracellular vesicles (AEVs) have not been systematically characterized, particularly for non-extremophilic archaea associated with the human host. Prior to this study, vesicle formation in these archaeal species had not been reported. This lack of data hinders our ability to differentiate archaeal contributions from bacterial ones within the gut ecosystem and to identify archaeal-specific mechanisms of host interaction. Investigating AEVs could reveal novel signaling pathways and bioactive molecules unique to archaea, potentially uncovering distinct roles in microbiome stability and host modulation that are not observed with bacterial EVs."

2. The authors concluded that adhesins/adhesin-like proteins are highly enriched in EVs,

but these seem to be localized on the bacterial surface or extracellularly. The enrichment of these proteins in EVs compared to the whole cell lysates is observed as a matter of course. Are these proteins more concentrated in the EVs as compared to the cell membrane fraction? Are they not present in extracellular supernatants (removed EVs)?

Additional experiments were performed to investigate the protein profile of cell membrane fractions, and the additional information was added to the manuscript:

"The proteomic profile of the archaeal cell membrane fraction (MF) was analyzed to investigate a potential enrichment in ALPs. Proteins were considered to be present based on a prevalence in at least 4 out of 5 replicates per strain (*M. smithii* ALI, *M. intestini*). Overall, approximately 4% of the proteins identified in the cell membrane fractions of *M. smithii* ALI and *M. intestini* were annotated as ALPs (total identified proteins = 1904, number of ALPs = 81). In contrast, ALPs accounted for 20% of the identified proteins in vesicles (total identified proteins = 229; ALPs = 46). Among the 46 ALPs identified in AEVs, 23 for *M. smithii* ALI and 41 for *M. intestini* were also found in their respective cell membranes. Additionally, the ALP profiles of *M. smithii* ALI and *M. intestini* shared 37 ALPs, while approximately 37% of their ALP profiles were distinct, highlighting species-specific differences (total ALPs *M. smithii* ALI: 57; total ALPs *M. intestini*: 60)."

Unfortunately, it was not possible to measure the extracellular supernatant as the concentration of proteins was below the detection limit for MS.

3. Similar concerns as above should be noted for the metabolomics results. Although glutamic acid and aspartic acid are enriched in EVs compared to the control medium, it remains unknown whether these amino acids are concentrated in EVs or whether they are also abundant in the supernatant. The authors need to investigate whether they are specifically present in EVs.

The control medium used for the comparison was non-cultured medium that had passed through the pipeline for vesicle isolation. Thus this question cannot be answered from the data obtained in this experiment.

However, an additional metabolomics experiment (NMR) was performed to analyze the production of aspartic and glutamic acid in the supernatant, during the growth of *M. smithii* and *Cand. M. intestini*.

Aspartic acid was not produced during growth (comparison 0h to 240h, both species; not detected in *Cand. M. intestini* samples, graph for *M. smithii* ALI is shown below).

However, glutamic acid was found to accumulate over time in *Cand. M. intestini* samples, but was not increased in *M. smithii* ALI samples. So it appears that at least aspartic acid is indeed a specific cargo for *Methanobrevibacter* vesicles.

As these experiments are difficult to integrate into the current body of the manuscript, due to the different set-up of the experiments, the results were not included in the newest version, but can be, in case the reviewer wishes.

4. To investigate the pro-inflammatory potential of archaeal EVs, the authors examined only the IL-8 cytokine response. For a comprehensive understanding of the immune response in epithelial cells, other cytokine production should also be examined. Although a certain concentration of EVs was examined in the current manuscript, it is necessary to investigate whether these are concentration-dependent responses.

According to this request, we performed additional experiments testing 23 more cytokines for HT-29 cells and also investigated a potential dose response. The results were added to the manuscript (chapter "AEVs induce various chemokines and cytokines in macrophages and

epithelial cells”).

5. The induction levels of IL-8 secretion are significantly different among species-derived EVs, but is this due to the presence of adhesion? The relationship between the presence of adhesion and the production of IL-8 is not clear at all.

Thank you for your comment. Unfortunately, the phrasing of the question is not entirely clear to us. If you are referring to the potential role of adhesins/ adhesin-like proteins (rather than adhesion), it is important to note that the function of adhesins in this context is not well understood. While adhesins have been implicated in mediating adhesion in certain cases, there is insufficient evidence to establish a direct association between adhesins and IL-8 production, particularly for the AEVs tested in our study.

As noted in our collaborator’s publication (Thapa et al., 2023, cited in the main text), IL-8 secretion appears to be highly species-dependent. The observed differences in induction levels are likely influenced by a range of factors, including small non-coding RNAs, proteins, lipoproteins, and other vesicle-associated components. Additional insights into these mechanisms can be found in the referenced publication.

Please note, that the entire section was re-written, due to a different experimental set-up performed to include additional cytokines/chemokines.

Other comments

6. Figure 1: Since only one vesicle of *M. intestini* is shown in the TEM images, several vesicles should be shown.

An additional image visualizing more vesicles was added (Figure 1F).

7. Please italicize the strain name appropriately in the article and in figures and legends. Also, the font size is too small in all figures, please make the figures larger. The number of biological replicates and measurements should be included in the figure legends.

We changed it accordingly, thank you.

8. Figure 2: It appears that the number of dots is different between strains. Does the number of dots on the graphs indicate the number of measurements or independent biological samples? In the legends (line 165), the criteria (threshold) for the values should be clearly stated.

The number of dots represent the technical and biological replicates, and the information was added to the figure legend. All single measurements are provided in Supplementary Tables, as referenced in the text. No threshold was used.

9. Line 202: Please include “Principal component analysis” in the text before using the abbreviation.

We changed it accordingly, thank you.

10. Figure 4 (B): What does "intensities" mean? Also, why is the order of the groups shown in the visualizations different between *M. smithii* and *M. intestini*? If there is no reason, please align them.

Intensity refers to relative abundance, this is now stated in the text.

The order of the groups follows the relative abundance of the shown proteins/ categories. This is now also mentioned in the Figure legend.

11. Line 248-251, Please include figure citations as appropriate in the text. For example, "Within a group of transport-associated proteins, we found substantial enrichment of a protein (representative: GUT_GENOME043902_01504) with an OPT (oligopeptide transporter) superfamily domain (Fig. 4B), which...".

The references to the Figures were included, as requested.

12. Fig. 4B: Show the intensities of proteins related to ①, ②, and ③ compared with those of whole cells, or heatmap like Fig. 4D for ①, ②, and ③. It is important to focus on proteins that are more abundant in EVs than in cells.

Indeed, these proteins are more abundant in AEVs than in cells. These proteins are indirectly included in the categories shown in Fig. 4C (Unknown cell wall-associated functions: DUF11 domain containing proteins, n=3), Proteolysis (Putative Peptidase C1, n=1), or Transport (Putative OPT family oligopeptide transporter, n=3) and all detailed information is provided in Supplementary Table 7.

An additional Supplementary Figure (S5) was created to show that these specific proteins are indeed enriched in the AEVs (see below).

13. Figure 5: Box and whisker plots are not appropriate because the authors only plotted three biological replicates.

This is a valid point, thank you! We changed the figure accordingly:

14. Line 291: In the case of "significantly changed", please indicate the calculation method and P-values.

This information was added.

15. The legend of Figure 6: please use EVs or AEVs consistently.

This was corrected throughout, thank you.

16. Figure 7: The bar graph must also include independent data (as shown by the dots). Why did the authors use "median and interquartile range", instead of "means \pm standard errors"?

The figures, and the according paragraph was completely changed due to additional experiments performed. Please see the updated version of the chapter "AEVs induce various chemokines and cytokines in macrophages and epithelial cells".

17. Line 331-333: The induction of IL-8 secretion by *M. intestini* EVs is not at the same level as by ETEC EVs.

The figures, and the according paragraph was completely changed due to additional experiments performed. Please see the updated version of the chapter "AEVs induce various chemokines and cytokines in macrophages and epithelial cells".

18. Line 368-369: The authors describe that "All vesicles were found to be taken up by human monocytes (Fig. 6) and to stimulate IL-8 secretion (Fig. 7, $P > 0.05$)", but only M. intestini EVs significantly induced IL-8 secretion. And what does this P value " $P > 0.05$ " mean?

The figures, and the according paragraph was completely changed due to additional experiments performed. Please see the updated version of the chapter "AEVs induce various chemokines and cytokines in macrophages and epithelial cells".

19. Line 531-532: Yeast vesicles seem to pass through filters.

We have taken extensive precautions to minimize the risk of transferring yeast vesicles to our AEV extracts. First, we filtered the yeast extract using 0.22 μm filters before utilizing it in any cultivation processes to ensure that large particles and vesicles are removed.

During the vesicle isolation, we implemented a rigorous three-step filtration process, followed by size exclusion chromatography, which further reduced the likelihood of yeast vesicle contamination.

Additionally, proteomics (which served as a final quality control) revealed that only a very minor proportion of proteins (30 out of 1672 proteins) could be traced back to yeast. This minimal presence strongly suggests that the risks of yeast vesicle contamination is negligible.

Reviewer #2:

Weinberger et al present a study on the capability of the archaeal genera Methanobrevibacter and Methanosphaera to produce extracellular vesicles (AEVs). The authors employed electron microscopy, proteomics and metabolomic analyses to characterize the properties of these AEVs. Furthermore, the authors explored the pro-inflammatory potential of these AEVs against human epithelial cell line.

While the study addresses an important area of archaeal biology, several aspects require further refinement. The evidence supporting the involvement of adhesin and adhesin-like proteins in archaea-surface and archaea-bacteria interactions, as well as the proposed connection between the identified glutamic acid and choline glycerophosphate with gut-brain signaling, appears to be insufficiently substantiated. I recommend that the authors consider tempering the confidence with which these conclusions are presented. Additionally, the overall clarity of the manuscript would benefit from significant improvements to ensure a coherent and strong logical flow throughout the text, and there are a few typo errors that needed to be checked and addressed. The resolution and readability of the figures should also be enhanced, as the current quality, particularly the small and blurred text within the figures, diminishes their effectiveness in conveying key points.

We are thankful for the reviewer's comments. We have toned down our assumptions on potential involvement in gut-brain signaling, as, of course, experiments on the causal effects were beyond this study, which aimed to report and initially characterized AEVs produced by human GIT-associated archaea.

Information included in the discussion:

“The metabolic profiling of AEVs revealed increased levels of aspartic and glutamic acid (Fig. 4), probably suggesting a potential link between AEVs and the gut-brain axis (as discussed in the results section). The possibility that AEVs could influence host neurological processes warrants further investigation. Given that BEVs are known to interact with their neighboring cells, cross the intestinal barrier, and enter the bloodstream, potentially reaching distant tissues such as the brain^{10,95}, it is plausible that AEVs exhibit similar properties. “

We have improved the clarity, checked for typos and improved the readability of the figures.

Major comments

1. Line 29 and Line 32. Please correct “*M. smithii*, *Cand. M.intestini*” & “*M. smithii* and *Cand. M.intestini*”

We changed it accordingly, thank you.

2. Line 32. Please clarify which the “unique traits” are being referred to and how they are relevant to the study’s findings.

This was now clarified, as follows: “While the size (~130 nm) and morphology of these AEVs were comparable to BEVs, proteomic and metabolomic analyses revealed unique traits. AEV proteins (n=229) of both *M. smithii* and *Cand. M. intestini* revealed a massive accumulation of adhesins/adhesin-like proteins, which may mediate AEV-bacteria and AEV-host interactions. Additionally, the AEVs contained free glutamic and aspartic acid and choline glycerophosphate, compounds which may be involved in gut-brain signaling.”

3. Line 33. It is not clear whether “n=229” represents the total proteins identified in the EVs of both *M. smithii* and *M. intestine*, or it refers to the proteins found in the EVs of *M. intestini*. There is no explanation for why only proteomic analyses were performed for the EVs of *Methanobrevibacter* and not for *Methanosphaera*.

We changed it accordingly to clarify, n=229 refers to proteins found in both AEVs (*M. smithii* and *M. intestini*) (see sentences copied to the above question).

We added an explanation on the proteomics analyses done only for *M. smithii* ALI and *Cand. M. intestini* in the Results section: “Proteomic analysis was conducted exclusively on *M. smithii* ALI and *M. intestini*, as our focus was on the two predominant archaeal species in the human gut, thereby excluding *M. smithii* GRAZ-2 and *M. stadtmanae* from this specific analysis.”

4. Line 37. Please clarify what IL-8 is and specify the purpose of using HT-29 cells, and explicitly state what the results involving IL-8 and HT-29 suggest.

IL-8 is a pro-inflammatory chemokine primarily involved in the immune response by attracting and activating neutrophils. It plays significant roles in various physiological and pathological processes, including inflammation, cancer progression, and angiogenesis.

We used HT-29 cells as those cells were recently used in a comparative study to assess the differential pro-inflammatory potency of bacterial EVs from gut bacteria (Thapa et al). As asked for in other comments, we performed additional immunoassays including 23 cytokines to assess a bigger immunostimulatory potential of AEVs.

Please note: The figures, and the according paragraph was completely changed due to additional experiments performed. Please see the updated version of the chapter "AEVs induce various chemokines and cytokines in macrophages and epithelial cells".

5. Line 64. The authors should provide more detailed information about explosive membrane vesicles, including where they are produced and how they differ from outer membrane vesicles. A clearer explanation of these distinctions would enhance the reader's understanding of their relevance to the study.

We have edited the manuscript to include this additional information:

"While OMVs are formed through blebbing, explosive membrane vesicles are generated via endolysin-induced cell lysis"

6. Line 89. Please specify what is meant by the "defensive function" of these vesicles. Defensive against what?

We have updated the respective paragraph: "It appears that in *Sulfolobus*, for example, vesicle formation is evolutionarily related to the eukaryotic endosomal sorting complexes required for transport (ESCRT) proteins used for the building of endosomes; however, other archaea, such as *Thermococcus* form vesicles but do lack the ESCRT complex, indicating a higher variety in vesicle formation mechanisms¹⁹. Vesicles formed by *Thermococcus* and other Thermococcales species serve multiple functions, primarily related to sulfur detoxification and genetic material transfer".

7. Line 93. I think "Trough" is meant to be "Through".

We changed it accordingly, thank you.

8. Line 114. Supps Figure 1 should be cited for "AEV biomass production and an isolation protocol"

We changed it accordingly, thank you.

9. Line 121-122. Did the authors observe any significant difference in the sizes of vesicle-like structures within the cells compared to the isolated purified vesicles? A proper quantitative analysis should be performed to substantiate any conclusions regarding size differences. From the figures, it appears that the vesicle-like structures within the cells are generally smaller than the isolated vesicles. Could the authors provide comments or additional analysis on this?

We have performed scanning electron microscopy of whole cells and transmission electron microscopy on ultrathin sections to visualize vesicle-like structures. However, it is important to note that these two methods require distinct sample preparation techniques, which can introduce variability in measurements. Consequently, direct comparisons of vesicle sizes between these methodologies should be made with caution.

While the vesicle-like structures within the cells appear smaller in the figures compared to the purified vesicles, this observation may be influenced by differences in sample preparation, imaging resolution, and the two-dimensional representation of intracellular vesicles.

A proper quantitative analysis would require a different experimental procedure, which was beyond the scope of the current study.

10. Line 123. The “culture media controls” is not clearly defined. Please clarify whether specific media was used to introduce vesicle-like structures and provide data demonstrating that no vesicles were observed in the culture media controls.

We have edited the manuscript to include this additional information (“non-cultured control medium that had passed through the pipeline for vesicle isolation”) and added an additional supplementary figure S4A, to show that no vesicles were observed in the culture media controls.

11. Line 129. From electron microscopy images, it would be more accurate to refer to these structures as “vesicle-like structures” unless further evidence explicitly confirms their identify as AEVs.

We changed it accordingly, thank you.

12. Line 137. Table 1 could be moved to the Supplementary Material. Additionally, please specify the number of biological replicates performed in the figure and table legends. The proper statistical analysis should be performed.

We changed it accordingly, added the required information, and moved the table to the supplementary material, thank you.

13. Line 142-143. References are required for “enterotoxigenic Escherichia coli (ETEC, ~120 nm)”, as well as “B. fragilis (~194 nm)”, respectively.

References are now added, thank you.

14. Line 145. Please specify the source of the data of ETEC (6.38E+11 particles/ml), and B. fragilis (8E+11 particles/ml).

We changed it accordingly, referring to supplementary table 2, thank you.

15. Line 146. The statement “The concentrations were reasonably consistent for all M.

smithii strains”.

All” should be revised to “both” if only two strains were analyzed. Please provide details on how this conclusion was reached. Proper statistical analysis should be performed to determine if there are significant differences or no differences between the strains.

We changed it accordingly and added the required information, thank you.

16. Line 153 “the lowest amount of lipids (~4.9 µg/10¹⁰ particles on average) for which strain?

This refers to *M. smithii* ALI, we added this information..

17. Line 163. The figure legend should be expanded to include more details. It should indicate the number of biological replicates, explain what the line inside the box represents, and describe the corresponding histogram and clarify the range of values depicted by the box.

This information was added, as requested.

18. Line 167. The table shows that the min and max values for lipid content are the same for *M. intestini*, *M. smithii* GRAZ-2, and *M. stadtmanae*. Please provide an explanation for this observation in the table legend.

The table was changed accordingly. This was also stated in the manuscript:

“Lipid content of AEVs was only partially within the range of the standard linoleic acid calibration (20–100 µg/ml). AEVs from *M. intestini* had the highest lipids content (~81.20 µg/10¹⁰ particles), whereas *M. smithii* ALI vesicles contained the lowest (~4.9 µg/10¹⁰ particles on average). As lipid content could only be measured in a limited number of samples (not all concentrations were within the standard range), no graphical display is shown, but all data are included in Supplementary Tables 1,2. Statistical analysis of lipid content revealed no significant differences ($p > 0.05$, Kruskal-Wallis test).”

19. Line 171. It is not clear why the protein profiles of EVs were analyzed only for *M. smithii* AL1 and *M. intestini*. Please clarify why the other two strains were not included in the analysis.

We included an explanation in the Results section: “Proteomic analysis was conducted exclusively on *M. smithii* ALI and *M. intestini*, as our focus was on the two predominant archaeal species in the human gut, thereby excluding *M. smithii* GRAZ-2 and *M. stadtmanae* from this specific analysis.”

20. Line 181. Supplementary Figure S2 is not readable. The figure legend should specify the origin of the 394 proteins mentioned. There is also no description of these 394 proteins in the main text.

Indeed, this supplementary Figure (S2) was providing additional, but not-relevant information, so the Figure is removed in the new version of the manuscript.

21. Line 182. Is there a specific reason for comparing BEVs from *B. thetaiotaomicron* in this context, rather than including BEVs from other species? Providing a rationale for this choice will help clarify the relevance of the comparison.

We removed the comparison as it is not relevant in this context.

22. Line 190. The Figure 3 could be moved to Supplementary Material. The figure is difficult to read due to the use of many abbreviations. Consider including species names in the diagrams, such as *M. smithii* AL1_EV. There appears to be sufficient space in the figure to for these labels to improve clarity. Having many abbreviations in the main text affects readability.

The figure was adjusted according to your suggestions and moved to the supplementary material (Supplementary figure S2). Thank you.

23. Line 208. The same comment as above. There is no need to have so many abbreviations.

We changed it accordingly, thank you.

24. Line 215. The description of “3+3 biological replicates” is not clear.

We changed it accordingly, to make it more clear, thank you; the entire Figure legend was rephrased:

“ Figure 3: **Mass spectrometry-based profiling of AEV proteomes.** (A) Principal component analysis (PCA) plot illustrating the protein profiles of AEVs and WCLs of *M. smithii* ALI and *M. intestini*, including only proteins detected in all three replicates per group (AEV *M. smithii* ALI, AEV *M. intestini*, WCL *M. smithii* ALI, WCL *M. intestini*). (B) Overlap of 229 proteins identified in the AEVs of *M. smithii* ALI (left, n=3 biological replicates) and *M. intestini* (right, n=3 biological replicates), visualized and organized by intensities/relative abundance (circle size) and functional categorization (see Supplementary Tables 5-7 for details). Data were visualized using RawGraphs⁶⁰ and InkScape⁶¹. (C) Bar chart displaying mean intensities/relative abundances of protein categories in AEVs and WCLs, based on proteins detected in all three biological replicates of both *M. smithii* ALI and *M. intestini* (n=229) (Supplementary Table 5-7). (D) Heatmap showing enrichment of 46 proteins annotated as adhesin/adhesion/Ig-like present in all six AEV extracts (three biological replicates each of AEV *M. smithii* ALI and AEV *M. intestini*) compared to the whole cell lysates based on relative abundances. Abbreviations: WCL, whole cell lysate; AEV, archaeal extracellular vesicles. “

25. Line 232-236. Strictly speaking, are these really domains (Invasin/intimin cell-adhesion fragments; Pectin_lyase_fold/virulence)? Does InterPro call these domains?

We changed “domain” to motif.

26. Line 237. "Bacterial proteins containing Ig-like domains exhibit a broad spectrum of functions,"

This sentence and the paragraph that follows seems to indicate that Ig-like "domains" and

pectin lyase folds are found in such diverse proteins that their function(s) in adhesin-like proteins cannot be inferred with any confidence.

This is correct; in bacteria these domains are diverse in function; for the archaea no information is available.

27. Line 240-242. Reference is required

Reference was added, thank you.

28. Line 243-245. Reference is required.

Reference was added, thank you.

29. L249-251. "OPT (oligopeptide transporter) superfamily domain, which in prokaryotes may contribute to iron-siderophore uptake, indicating a potential role in iron binding"

The meaning here isn't entirely clear. Is this OPT predicted to import oligopeptides or complexed iron or both?

It seems that both possibilities exist, according to the reference cited. We changed the sentence to make it clearer: "In general, OPT transporters are known for oligopeptide uptake but can also facilitate the transport of iron-siderophore complexes⁶⁶, indicating a potential role in iron uptake."

30. L260-262. Is this free glutamic and aspartic acid, or part of proteins/polypeptides contained in the EV? This is not clear.

Both amino acids are free amino acids and not part of polypeptides.

This was now added to the manuscript: "Strikingly, the AEVs of *M. intestini* revealed a significantly increased content of free glutamic and aspartic acid"

31. Line 262. It is not clear whether "a substantial Log₂ Fold change (FC)" refers to changes in abundance or another measurement. The "background samples" is not clearly defined. Please also explain how the conclusion that "these amino acids are important cargos for both species" was reached.

This part was rephrased accordingly, and a reference to Fig. 5 is given, where more details are provided: "Similarly, the AEVs of *M. smithii* ALI revealed a substantial (but not significant) increase of these amino acids compared to the negative control (incubated, empty medium) (Fig. 4)."

32. Line 301. Please specify whether DiO was used to label the membrane of AEVs.

We changed it accordingly, thank you.

33. Line 309 and 313. Please specify which microscope was used to capture these images in the figure legend. Panel E shows an orthogonal slice image rather than a representative

Z-stack. Please label the X-Y, Y-Z, and X-Z planes in the figure. To better display the results, please place Z-slice images at different Z-depths on the right side.

We have edited the manuscript to include this additional information and images.

34. Line 343. How did the authors ensure that the particle amounts of AEVs were equal? Additionally, could the size of the AEVs affect the results? Please address these concerns to clarify how the study controls for these variables?

All batches of AEVs were measured via NTA to investigate the concentrations in particles/ml. Overall, the sizes of the AEVs were relatively similar (and variability was also observed in BEVs), so we do not expect that the variations would affect the results.

35. Line 371-373. "Methanobrevibacter species rely on syntrophic bacterial partners that provide small organic compounds like H₂ (or formate) and CO₂ for methanogenesis"
It might not be correct to refer to H₂ as organic, even if (as here) it has a biotic origin.

This is correct, organic was removed.

36. Line 383. "From the bacterial kingdom"--domain, not kingdom.

We changed it accordingly, thank you.

37. Line L385 "Considering that adhesins are highly enriched in AEVs"

The proteins mentioned earlier are not demonstrated to be adhesins, but adhesin-like proteins. A function in adhesion has yet to be determined, and the presence of Invasin/intimin cell-adhesion fragments and Pectin lyase fold does not necessarily point to a function in adhesion.

This is correct, we changed to "adhesins/adhesin-like proteins", and rephrased corresponding sentences in the manuscript.

38. L388-389 "In *Methanobrevibacter ruminantium*, a prevalent *Methanobrevibacter* species in ruminants, 5% of the genome encodes adhesins"

The functions of almost all of these proteins as adhesins has yet to be demonstrated. Mru_1499 is exceptional in that it has been investigated experimentally.

Correct. We changed the wording as follows: "In *Methanobrevibacter ruminantium*, a prevalent *Methanobrevibacter* species in ruminants, 5% of its genome is predicted to encode putative adhesins or adhesin-like proteins"

Reviewer #3:

The manuscript focuses on the characterization of extracellular vesicles (EVs) isolated from archaeal strains that inhabit the human gut. In addition to the proteomic and metabolic profiling of the isolated vesicles, authors conduct functional studies in macrophages and

intestinal epithelial cell lines.

The study of archaeal extracellular vesicles (AEVs) is a relevant research area in the context of the gut microbiota. In the last years, research on the gut microbiome has been almost exclusively focused on bacteria. In this context, studies on other microbiota components such as viruses, fungi and archaea are needed to understand their contribution to intestinal homeostasis and human health.

Major comments

1. The first paragraph of the results section (lines 110-116) is just a summary of the following result subsections. It could be removed, and the information introduced into each specific subsection.

We have edited the manuscript accordingly.

2. Figure 1 shows EVs inside the cells. How can authors prove that these are intracellular vesicles that originate the EVs? Which is the mechanism of EV formation in these strains? Did the proteomic analysis identify ESCRT proteins in the EVs? This information may shed light on the potential mechanism of vesicle biogenesis in intestinal archaeal strains.

Thank you for your comments. Figure 1 also includes thin sections of the cells, showing vesicle-like structures inside the cells. Regarding the vesicle-like structures inside the cells shown in Figure 1, we are at the early stages of EV characterization, and thus, the mechanism of vesicle formation remains to be fully investigated.

As for the potential involvement of ESCRT proteins in vesicle biogenesis, our proteomic analysis did not detect any ESCRT-related proteins. However, we acknowledge that the absence of these proteins in our data does not necessarily imply their absence in the vesicles; it may reflect limitations in protein identification or annotation within the current dataset.

3. Figure 1 and Figure 3: Lettering and legends are illegible even in the original figures (big size).

We have edited the figures to make them more readable.

4. Table 1. There is a great variability in protein and DNA concentration data from different batches of EVs isolated from a single strain (as an example, protein concentration ranges from 0.09 $\mu\text{g}/10^{10}$ particles to 180 $\mu\text{g}/10^{10}$ particles). In the discussion (lines 359-366), the authors point to growth conditions (culture phase and medium composition) as main factors influencing the vesicle size, amount and cargo. In this sense, EV samples collected from cultures carried out with the same strain, medium, and conditions should be more homogeneous. This is relevant since data presented in table 1 are normalized to 10^{10} particles. Normalization by particle number should yield more similar values.

Thank you for raising this important point. We have included detailed information on technical and biological replicates in the respective Supplementary Tables and have also updated the data presented in Fig. 2 for greater clarity.

While the overall variation e.g. in protein concentrations appears broad, the reasons for this variability remain unclear and may involve factors beyond the culture conditions we controlled for, such as intrinsic strain-specific differences or subtle, unmeasured environmental influences. Notably, vesicles from *M. smithii ALI* and *M. intestini* exhibited less variability in their protein and DNA concentrations. This observation also supported our decision in selecting these strains for more detailed analyses.

5. Lines 181-183: Authors compare the number of proteins identified in archaeal EVs with the number of proteins identified in BEVs from *B. thetaiotaomicron* and conclude that the number of proteins in archaeal EVs is lower. Which is the reasoning for this comparison? The number of vesicular proteins identified depends on the methodology used, data curation, the producer strain, and the growth conditions. In reference 57, the EV analyzed were collected from in vitro BHI cultures and from in vivo samples. Moreover, I am not sure whether the 2,047 proteins indicated in the manuscript (lines 181-182) are indeed proteins identified in *B. thetaiotaomicron* EVs or in the parental cells after data curation (see reference 57, page 11, section “Proteomics Data Curation”).

We agree with your comment and have removed the comparison between AEVs and BEVs from *B. thetaiotaomicron*, as it did not provide relevant or meaningful context to our study. Our initial intention was to offer readers some comparative insight from BEV studies; however, given that this point was raised by multiple reviewers, we have decided to omit the comparison completely.

6. I am not expert in metabolomics, but the analysis has identified very few metabolites. Please discuss.

The limited number of identified metabolites is primarily due to the small amount of biomass available for analysis. Additionally, we applied stringent identification criteria, relying exclusively on metabolites supported by our internal standard database (“Annotation of the compounds was done through searching against our internal mass list database generated with authentic standard solutions (highest confidence level)”). This approach ensures a high level of confidence in our metabolite identifications but naturally results in a lower overall number of identified metabolites.

While this may appear limiting, we believe it is essential to prioritize accuracy and reliability in metabolite identification, and we have therefore focused on those metabolites that could be characterized with the highest level of confidence.

7. The metabolomic profiles indicate that EVs from *M. smithii ALI* and *M. intestine* are enriched in glutamate and aspartate. Did authors check whether the high amount of these amino acids in EVs correlate with proteomic data showing the presence of metabolic enzymes that catalyze their production (such as asparaginase) as vesicle cargo?

Indeed, asparaginase was indeed detected in the whole-cell lysates of *M. smithii ALI* and *M. intestini*, but it was only identified once in an AEV extract from *M. smithii ALI*. Due to its

limited detection in AEVs, it did not become a focal point of this manuscript, as we primarily concentrated on proteins consistently identified in AEVs across replicates.

However, in the AEVs (see Supplementary Table 5), we identified a protein from the succinylglutamate desuccinylase/aspartoacylase family, which may play a role in glutamate formation from succinylglutamate. This finding could suggest a metabolic link to the enrichment of glutamate in the vesicles, though further experimental validation would be necessary to confirm this connection.

8. Why authors study internalization of AEVs in THP1-derived macrophages but analyze their pro-inflammatory effect (IL-8 secretion) in HT29 epithelial cells?

Macrophages are phagocytic cells, so they can take up vesicles by means of this activity, whereas intestinal epithelial cells, which are non-phagocytic cells, usually internalize EVs through endocytosis. Immunofluorescence microscopy analysis of labelled vesicles in HT29 cells would provide knowledge on the internalization pathway of AEVs in epithelial cells.

Concerning the inflammatory response, why not analyze cytokine secretion by macrophages in response to AEVs? In fact, macrophages are immune cells.

Internalization studies in HT29 cells and cytokine analysis in THP1 derived macrophages would complete the functional part of the manuscript.

Additional experiments were performed to investigate the cytokine analysis in THP1 derived macrophages and included more cytokines in the cytokine analysis of HT-29 cells to get a better overview.

Our goal was to investigate the interactions between AEVs and immune cells and also epithelial cells to see potential pro-inflammatory effects on those cells indicated by AEVs.

Differently to macrophages, epithelial cells are not expected to take up vesicles (at least not in a substantial amount, as a whole), so we did not perform experiments on the internalization of AEVs in epithelial cells.

9. Lines 300, 678: what does the letter b in THP1-b mean?

THP1-b is the abbreviation for THP1-Blue™ cells which are derived from THP-1 monocytes, to monitor NF-κB signal transduction. We used this specific version of THP-1 cells as we also tested the NF-κB signal transduction (not shown in the manuscript). The explanation was added to the manuscript, thank you.

10. Please provide details on the PMA protocol used to differentiate THP1 monocytes into macrophages (incubation time with PMA and days of rest) in the confocal microscopy section (line 679).

We have edited the manuscript to include this additional information.

11. Lines 531-532: Which filter size have authors used to eliminate the vesicles and other residues from the yeast extract used for medium preparation? Vesicles smaller than 200 nm are not eliminated using a 0.22 μm filter.

To minimize the risk of transferring yeast vesicles or other contaminants into our AEV extracts, we filtered the yeast extract with 0.22 µm filters prior to its use in cultivation processes. While we acknowledge that vesicles smaller than 200 nm may not be eliminated by this step, additional precautions were taken during the vesicle isolation process. Specifically, we employed a rigorous three-step filtration process followed by size exclusion chromatography, which further reduces the potential for contamination by yeast vesicles.

As a final quality control, our proteomic analysis revealed that only a very minor fraction of proteins (30 out of 1,672 identified proteins) could be attributed to yeast. This minimal presence strongly supports the conclusion that the risk of yeast vesicle contamination in our AEV samples is negligible.

12. Line 358: "...although the particle count was substantially lower for archaea (Fig 2., Table 1)". Table 1 and Figure 2 do not present data for bacterial EVs.

This is now corrected accordingly, pointing to Supplementary Table 2.

Minor comments

13. Line 11 and abstract: the strain names should be in italics.

We changed it accordingly, thank you.

14. Line 447: In the following, we will use the abbreviation *M. intestini* instead of *Candidatus M. intestine*. Since the Methods section is at the end of the manuscript (after discussion) this information should be provided the first time the strain appears (in the results section) to help the reader.

We changed it accordingly, thank you. This information was included in the Results section.

15. Lines 455-456: the sentence lack the verb.

We corrected it accordingly, thank you.

16. Line 369: ...and stimulate IL-8 secretion (Fig 7...). Please, add in intestinal epithelial cells after secretion.

We changed it accordingly, thank you. Please note the overall changes associated with the additional experiments performed as explained in this paragraph: "AEVs induce various chemokines and cytokines in macrophages and epithelial cells"

17. Line 378: communication vehicles or molecules? The sentence refers to adhesins...

We changed to "communication molecules", thank you.

Reviewer #4:

The manuscript entitled "Proteomic and metabolomic profiling of archaeal extracellular vesicles from the human gut" is a strong foundational study analyzing the vesicle content and basic immune stimulatory ability of EVs derived from two biologically important archaeal species. The authors present a thorough and detailed analysis of EVs produced by archaeal species commonly present in the gastrointestinal tract. The importance of archaea in the overall function of the microbiome and health of the host underpins the significance of the presented work as the mechanisms through which these microbes communicate with each other and the host are poorly understood. The robust nature of the vesicle data presented and the stringent thresholds for vesicle analysis are among the strengths of this manuscript. While interesting, the biological significance of the metabolite data is unclear given that these bacteria are grown in nutritive media under laboratory conditions. However, the robust nature of the metabolome analysis does provide a strong foundation of information and a starting point for metabolomic analysis of host-derived vesicles. In contrast to the vesicle analysis data, the biological impact and cell response studies were not nearly of the same caliber. The authors give brief methods for cell viability assays (which admittedly are standard) but neglect to mention which AEV or BEVs are specifically tested with the cells and also do not mention what controls were used. Considering cell viability impacts immune responses, this data should be shown for all EVs used in IL-8 experiments. Dye only controls were also missing for EV entry experiments. In addition, the evaluation of only one cytokine as an indicator of immune stimulation is disappointing, particularly in light of the very thorough nature of authors vesicle analysis. Overall, the author's characterization of the vesicles is superb but there are concerns regarding the analysis of vesicle interaction with host cells.

Thank you! We have included additional experiments as outlined in "AEVs induce various chemokines and cytokines in macrophages and epithelial cells" as requested. Limitations for the metabolomic analyses were now included in a separate limitations paragraph in the discussion part.

Major comments

1. Line 93. Should the last word be "through"?

We changed it accordingly, thank you.

2. Line 145. Include representative NTA graphs as a supplemental figure.

A representative NTA graph was added to the supplementary information (Fig. S4).

3. Line 159. The authors do not mention in the methods if EVs were DNase or RNase treated prior to measuring DNA and RNA, respectively. If this was done, the methods used should be included in the methods sections. Extraneous, unencapsulated nucleic acids are commonly detected in EV preps and the lack of DNase/RNase treatment can inflate nucleic acid measurements. If these treatments were not done, experiments should be repeated or that caveat included in the results description and discussion.

Indeed, DNase/RNase was not used prior the measurements. This information was added to the methods and discussion part, thank you.

“While this study provides substantial insights into the role of AEVs in microbial communication and host interactions, several limitations must be acknowledged. The DNA and RNA content of vesicles was measured without prior DNase or RNase treatment, meaning the results may include measurements of surface-attached nucleic acids. Further, the analyzed AEVs were derived from monocultures grown in artificial growth media, which may not fully represent their natural state in the gastrointestinal tract. As a result, findings from e.g. metabolomics and proteomics should be validated using host-isolated vesicles, though archaeal vesicles are likely underrepresented in such samples. Additionally, while biological replicates were used, natural fluctuations in vesicle cargo composition were observed, likely due to vesicle heterogeneity. Different vesicle subtypes may carry distinct cargo, leading to varied biological effects, depending on the targeted microbial or host cells. Moreover, the isolation process itself might impact the retrieval of different vesicle subtypes.”

4. Figures 2 and 4. The font size for axis legends and labels are impossible to read and need to be increased in size.

We changed it accordingly, thank you.

5. Line 174. The statement to "refer to materials and methods section" is unnecessary since that is always where such details are located. The statement should be removed.

We changed it accordingly, thank you.

6. Line 229. A reference to the publication showing that "rumen methanogens account for up to 5% of all genes" should be included.

This reference was added accordingly.

Leahy, S.C., Kelly, W.J., Altermann, E., Ronimus, R.S., Yeoman, C.J., Pacheco, D.M., *et al.* (2010) The genome sequence of the rumen methanogen *Methanobrevibacter ruminantium* reveals new possibilities for controlling ruminant methane emissions. *PLoS ONE* 5: e8926.

7. Line 261. Since the bacteria are grown in vitro in nutritive media, it is unclear what the biological relevance of detected, packaged metabolites will be. Since substrates will be different in the gut, it stands to reason that packaged metabolites might also be different. Authors should address this in the discussion.

We have added a limitation paragraph in the discussion (last paragraph), to address these and other limitations.

8. Line 302. Were dye controls used in this experiment? Dye aggregates regularly form with Dio and dye only controls that have been processed in parallel with (or at least in the same manner as) the OMVs and then applied to cells in parallel should be included to ensure that the signal observed is vesicles and not dye aggregates which are also endocytosed by macrophages. If this was not done, dye only controls should be tested to confirm that it is

actually AEVs being taken up by the cells and not dye aggregates, particularly for larger fluorescent spots as dyed vesicles tend to produce punctate signals.

We changed it accordingly, thank you.

We are very aware of the limitations of using lipophilic dyes, such as the formation of dye micelles and aggregates. Free dye controls were not included in this experiment as we have investigated in detail free dye uptake, including DiO, in our previous studies using labelled bacterial extracellular vesicles for both in vitro and in vivo assays (e.g. Jones et al, Front Microbiol. 2020; Modasia et al, J Extracell Biol. 2023). We included dye-only controls as part of these studies and found no unbound, free dye remains in the PBS following washing of the BEVs using centrifugal filters. The same protocol was used for this publication with A-EVs We discuss the limitations of using lipophilic dyes such as DiO in a recent publication (Jones et al, Methods Mol Biol. 2024). In addition, if DiO is directly added to cell cultures in vitro to simulate free dye, the majority of dye can be seen to incorporate into the cellular membrane as well as intracellular puncta (unpublished). We have edited the manuscript materials and methods to include more detail for the DiO labelling method.

9. Line 398. It is unclear why this sentence is set aside from the paragraph above. It's grammatically incorrect to have a standalone sentence, so it should be combined with the either the preceding or following paragraph.

It is now combined with the previous paragraph.

10. Line 404. which responses by HT-29? Assumed IL-8, but that should be specified since figures 6 and 7 are previously mentioned in the paragraph.

We changed it accordingly, thank you.

11. Line 408. The limitations of drawing conclusions from metabolomic data generated from EVs produced in culture media should be acknowledged in this paragraph.

This aspect is now included in the discussion (last paragraph on limitations).

12. Line 460. Could the authors elaborate on "prepared earlier"? How much earlier? There is a separate description of the methods used for BEV isolation, so the inclusion of this sentence in this methods section seems out of place and it's purpose in this location unclear.

This was corrected.

13. Line 517. If they were not processed immediately, how long were they stored at 4C? Considering long term storage results in degradation of EVs, the max amount of storage time is an important experimental consideration and should be included.

Additional information was added to the manuscript. "For long-term storage, vesicles were stored at -20°C. Vesicles were freshly prepared for each experiment. "

14. Line 530. Were protease inhibitors or any other diluents other than PBS used for isolation or final resuspension of AEVs?

No, protease inhibitors or any other diluents besides PBS were not used during the isolation or final resuspension of AEVs.

15. Line 666. What AEVs were used in cell viability studies? What controls were used? Cell viability data should be included in supplemental section, particularly considering that cell viability can impact immune responses and differences in IL-8 stimulation are seen between AEVs and BEVs.

Additional information and data was added to the manuscript and Supplementary Material, thank you.

16. Line 701. How many EVs were actually applied to the HT-29 cells? The concentration is given but not the volume, which is needed for anyone wishing to reproduce these experiments. It has been shown that EV number matters when evaluating immune responses by cells, so this information is important for evaluation of the data.

We have now extended this information in the text and also in the Figure legend (e.g. Fig. 6). "EV dose is indicated (A) 10^8 particles/ml cell culture medium and (B) 10^9 particles/ml cell culture medium on the bottom, as well as the vesicle origins."

17. Figure 7. The authors state $n=6$ but it is unclear based on the information included in the methods if that is 6 experimental replicates, 3 experimental replicates with 2 technical replicates, or some other technical/biological replicate combination. This should be clearly stated so the robustness of the result and analysis can be determined.

Figure 7 (now Fig. 6) was completely changed due to additional experiments performed.

The detailed description of experimental methods was excellent, particularly those associated with vesicle analysis. A common problem in the field is lack of this information which hampers the ability to replicate methods between laboratories.

Thank you very much for your comment, we highly appreciate it!

Reviewers Comments and Author Responses

We appreciate the time of all four reviewers and are thankful for their helpful comments and valid corrections.

Please find our responses below.

Reviewer #1:

Although the authors answered questions carefully, the current manuscript is still unlikely to be accepted for publication in Nature Communications.

Although I addressed that “Although proteomic and metabolomic analyses were performed, a detailed analysis of the factors contained in EVs was not performed.”, this issue has not yet been resolved in the revised manuscript. In particular, the title only states that the manuscript provides proteomic and metabolomic profiling, without clearly conveying what is novel and significant. This may be due to the lack of conclusive evidence identifying the key factors in the current research results. The authors rely solely on OMICS data and do not present definitive findings to support strong conclusions. If Adhesin or specific amino acids in EVs definitely play a critical role in gut cell linkage, further experiments will be needed to confirm this. There is a complete lack of data in the current manuscript to indicate what the impact would be in the absence (or increase or decrease) of adhesin or the specific amino acids. The title should specify which factors are important, rather than simply stating "Proteomic and Metabolomic Profiling."

We are thankful for the comment and agree that additional experiments are needed to assess the mechanisms of signal transfer, or to identify additional factors transported by the AEVs. We would like to stress, that *Methanobrevibacter* AEVs had not been described before, and this is the first assessment of their composition. Although the level of analysis is not comparable to the level of knowledge in bacterial EVs we still think that the data retrieved is meaningful and guiding the way for additional research.

With regard to each comment,

1. The newly added text aptly describes the importance of this study.
2. The authors explained that adhesion is highly contained in EVs.
3. The shown data is important and it is better to add into the supplemental results.
4. The authors responded appropriately to the comments and the data obtained is important.
5. I understand that IL-8 production is not determined by the presence of adhesin.

For other minor comments, authors correctly answered and revised the manuscripts.

Thank you!

Reviewer #3:

The authors' efforts to address the reviewers' requirements were remarkable. The inclusion of new data and experiments has significantly strengthened the manuscript. Notably, the proteomic analysis of archaeal cell membrane fractions, which confirms the enrichment of adhesin-like proteins in the corresponding AEVs (as suggested by Reviewer 1), and the multiplex analysis of cytokines secreted by THP-1 macrophages and HT-29 cells (as recommended by multiple reviewers) have substantially enhanced the study.

The authors have addressed all my comments and suggestions from the initial manuscript review process, except for the internalization of AEVs in HT29 cells (item # 8). I disagree with the explanation provided in their response letter: *“Differentially to macrophages, epithelial cells are not expected to take up vesicles (at least not in a substantial amount, as a whole), so we did not perform experiments on the internalization of EAVs in epithelial cells”*.

Numerous studies with bacterial EVs (BEVs), either from Gram-negative and Gram-positive bacteria, have shown that epithelial cells efficiently internalize BEVs through endocytosis. Various endocytic pathways contribute to BEV uptake, depending on the vesicle size and cargo, including clathrin-mediated endocytosis, lipid raft- and caveolin-dependent endocytosis, micropinocytosis and even direct fusion membrane. The use of specific inhibitors targeting these pathways have proven invaluable in elucidating the mechanisms of BEV internalization. For further details, see for instance the review with doi: 10.1111/cmi.12655 and the article with doi: 10.1038/s41598-020-78920-z).

If uptake experiments in HT29 cells are not conducted, the authors should state in the manuscript that further studies are needed to characterize the internalization pathway in intestinal epithelial cells. The different profile of adhesins in AEVs, depending on the producer strain, may influence or determine the entry route.

Thank you, we added this statement to the discussion: *“However, further studies are needed to explore the uptake in other cell types, such as epithelial cells, and to assess the impact of different AEV types and compositions on the mechanisms of entry into host cells.”*

Minor points:

Line 139: “... and attached to cells (Fig.1: a, e, **K**, l)...” The “K” should be “l”.

Corrected.

Line 204 (Figure 2 legend): “...A one-way ANOVA ($p=0.007$) followed by Tukey’s HSD post hoc test revealed”. I believe that ($p=0.007$) should be removed here.

Corrected.